# Impact of Hurricane Irma on Coral Reef Sediment Redistribution at Looe Key Reef, Florida, USA

Kimberly K. Yates[1], Zachery Fehr[2], Selena Johnson[1], David Zawada[1]

[1]U.S. Geological Survey, St. Petersburg, FL, 33701, United States
[2]Cherokee Nation System Solutions, Tulsa, OK, 74166, United States, Contractor to the U.S. Geological Survey

*Correspondence to*: Kimberly K. Yates (kyates@usgs.gov)

**Abstract.** Understanding event-driven sediment transport in coral reef environments is essential to assessing impacts to reef species, habitats, restoration, and mitigation, yet there remains a global knowledge gap due to limited quantitative studies. Hurricane Irma made landfall in the Lower Florida Keys with sustained 209 km $h^{-1}$ winds and greater than 8 m waves on 10 September 2017, directly impacting the Florida Reef Tract (FRT), and providing an opportunity to perform a unique comprehensive, quantitative assessment of its impact on coral reef structure and sediment redistribution. We used lidar and multibeam derived digital elevation models (DEMs) collected before and after the passing of Hurricane Irma over a 15.98 $km^2$ area along the Lower FRT including Looe Key Reef to quantify changes in seafloor elevation, volume, and structure due to storm impacts. Elevation change was calculated at over 4-million point-locations across 10 habitat types within this study area for two time periods using data collected from 1) approximately one year before the passing of Irma and three to six months following the storm's impact, and 2) from three to six months after, and up to 16.5 months after, the storm. Elevation-change data were then used to generate Triangulated Irregular Network (TIN) models in ArcMap to calculate changes in seafloor volume during each time-period. Our results indicate that Hurricane Irma was primarily a depositional event that increased mean seafloor elevation and volume at this study site by 0.34 m and up to 5.4 $Mm^3$, respectively. Sediment was transported primarily west-southwest (WSW) and downslope modifying geomorphic seafloor features including the migration of sand waves and rubble fields, formation of scour marks in shallow seagrass habitat, and burial of seagrass and coral-dominated habitat. Approximately 16.5 months after Hurricane Irma (during a 13-month period between 2017 and 2019), net erosion was observed across all habitats with mean elevation-change of -0.15 m and net volume change up to -2.46 $Mm^3$. Rates of elevation change during this post-storm period were one to two orders of magnitude greater than decadal and multi-decadal rates of change in the same location, and changes showed erosion of approximately 50% of sediment deposited during the storm event as seafloor sediment distribution began to re-equilibrate to non-storm sea state conditions. Our results suggest higher resolution elevation-change data collected over seasonal and annual time periods could enhance characterization and understanding of short-term and long-term rates and processes of seafloor change and help guide post-storm recovery and restoration of benthic habitats in topographically complex coral reef systems.

## 1 Introduction

Coral reefs provide a variety of services to coastal communities including protection from coastal hazards such as storms, waves, and erosion (Ferrario et al., 2014; Storlazzi et al., 2021); socioeconomic benefits such as fisheries, recreation, and tourism (Moberg and Folk, 1999; Hall et al. 2020); and they support numerous habitats and diverse marine species (Knowlton, 2020). Socioeconomic benefits of Florida reefs have an estimated value of over 8 billion dollars a year, supporting 39,000 South Florida jobs and 70,400 total jobs, with at least 2.9 billion dollars contributing directly to the local economy (Krediet et al., 2009; Gorstein et al., 2016, Towle et al., 2020). Benthic communities of the Florida Reef Tract (FRT) have been degrading for the past several decades. Coral coverage has declined across the Caribbean and Florida reefs by more than 50% since the 1970's due to coral disease and bleaching (Porter et al., 2001; Patterson et al., 2002; Williams and Miller, 2012; Joyner et al., 2015; Walker et al., 2018), pollution and overfishing (Littler et al., 1986; Lapointe & Clark, 1992; and Hughes 1994), and mass-mortality of macroalgal grazers (e.g., Lessios et al. 1983). Progression of climate change has increased thermal stress, coral bleaching and disease, ocean acidification, and corallivory (predation of corals) (Wilkinson 1996; Mumby et al., 2006; Brandt and McManus, 2009; Soto et al., 2011; Kuffner et al., 2015; Randall and van Woesik, 2015; Muehllehner et al., 2016; Hughes et al., 2017; Rice et al., 2019). These multiple stressors and increased storm occurrences have caused a shift from stony-coral-dominated reefs to macroalgae and octocoral dominated reefs (Bohsnack 1983; Hughs, 1994; Knowlton, 1992; Miller et al., 2002; Norstrom et al., 2009; Bruno et al., 2009; Ruzicka et al., 2013 and Jackson et al., 2014). Coral coverage has been reported at less than 7% along the Florida Keys Reef Tract and less than 3% along the northern FRT in recent years (Jackson et al., 2014; Walton et al, 2018; Knowlton, 2020); and many of Florida's reefs are in a net erosional state (Yates et al., 2017; Morris et al, 2022). Additionally, seagrass has been decreasing in coverage since early *Thalassia testudinum* die-offs in 1987 and more contemporary die-offs in 2015 following storm events and water quality variations (Hall et al., 2016).

Multi-decadal seafloor elevation-change analyses along the FRT indicate that degradation of coral reefs and surrounding seafloor habitats has led to substantial erosion and loss of elevation from the 1930's to 2002 and increased water depths to levels not expected until near the year 2100 (Yates et al., 2017). Projected socioeconomic impacts due to continued FRT coral reef degradation and loss of seafloor elevation estimate increases of flooding risk from storms and coastal inundation to more than 7,300 people and $823.6 million (2010 U.S. dollars, USD) in direct and indirect damage to housing, buildings, and businesses, annually (Storlazzi et al., 2021). Storm frequency and strength are projected to increase as sea-surface temperatures and atmospheric energy increase due to climate change and global warming (Elsner et al., 2008, Bhatia et al., 2019; Knutson et al., 2020). While advances have been made in understanding long-term change in seafloor elevation and structure and its potential socioeconomic consequences, understanding the effects of event-driven changes to seafloor geomorphology due to storms remains a major knowledge gap.

Major tropical storms persistently impact the state of Florida with historical hurricane impacts estimated to have caused more than $450 billion dollars of damage across the state from the early 1900's to 2007 (Malmstadt et al., 2009). The Middle to Lower Florida Keys (from Islamorada to Key West) has been impacted by 15 major hurricane landfall events (Category 3 through 5) and numerous tropical storm and Category 1 and 2 hurricanes from the early 1900's to 2022 (NOAA, 2022a). Hurricane Irma made landfall at Cudjoe Key in the Lower Florida Keys after passing directly over Looe Key Reef on 10 September 2017 as a category 4 hurricane with maximum wind speeds of 213 km $h^{-1}$ (115 kts) (Cangialosi et al., 2021) and significant wave heights of approximately 14 m a few kilometers offshore of the Florida Keys (Xian et al., 2018, Fig. 1a, b). Satellite imagery showed extensive sediment plumes throughout South Florida and the FRT caused by sediment resuspension and transport during the storm (Fig. 1c, d). The storm damaged up to 75% of buildings near its landfall point and caused approximately 50 billion USD of wind and water damage across the state of Florida (Xian et al., 2018; Cangialosi et al., 2021; NOAA, 2022b). Prior to Hurricane Irma, the most recent, direct impact to Looe Key Reef from a tropical storm was in 2008 during Tropical Storm Fay (NOAA, 2022a).

Numerous rapid assessments of seafloor habitats were conducted along the FRT in the weeks following Hurricane Irma. Diver-based surveys of coral reefs at 57 locations along the FRT by the National Oceanic and Atmospheric Administration showed highest levels of damage in the Middle to Lower Keys including dislodged and fractured corals, clogged and damaged sponges, heavy sedimentation, burial of corals, displaced rubble and sand, reef erosion, fractured substrate, and marine debris; 14% of sites showed severe impact, 33% showed moderate impact, and 53% showed minimal impact (Viehman et al., 2018). Looe Key Reef, located near the hurricane landfall location, showed more than 26% prevalence of hurricane-impacted corals (Florida Resilience Program, 2017). Similar surveys along the northern FRT from Key Biscayne north showed from approximately 5% to 17% of 62 sites with impacts to corals including dislodged and buried colonies, and at least one site with slabs of hardbottom 2 to 5 m in size fractured and displaced several meters (Walker, 2018). Analyses of long-term monitoring-transect data at 40 sites throughout the Florida Keys National Marine Sanctuary (FKNMS) showed instantaneous losses in seagrass and calcareous green macroalgae density after the storm passed, particularly in the Lower Florida Keys near where Hurricane Irma made landfall (Wilson et al., 2020). Additionally, several locations showed moderate burial of seagrass with up to 5 to 10 cm of sand, while other locations showed heavy erosion or moderate seagrass canopy thinning (Wilson et al., 2020). Reef Visual Census (RVC) surveys including structure from motion (SfM) habitat photogrammetry at sites in the Lower Florida Keys from February 2017 to December 2018 showed a 30% decrease in macroalgae at the Looe Key Sanctuary Preservation Area (SPA) and a 30% increase at the Looe Key Special Use Area (SPU) post Irma; while both Looe Key locations showed a 10% decrease in live coral cover and a 20% increase in octocoral cover (Simmons et al., 2022). Comparison of restored (outplant) coral survival rates at two fore reef and two patch reef sites near Tavernier Key in the Upper Florida Keys showed approximately 85% outplant survival at all locations prior to the passage of Hurricane Irma; however, no outplants survived at the fore reef sites and only 51% of outplants survived at the patch reef sites post-Irma, the difference likely due to protection of the patch reefs from dissipation of wave energy by the reef crest (Lohr et al., 2020). Examination of *Diadema antillarum* sea urchins (a

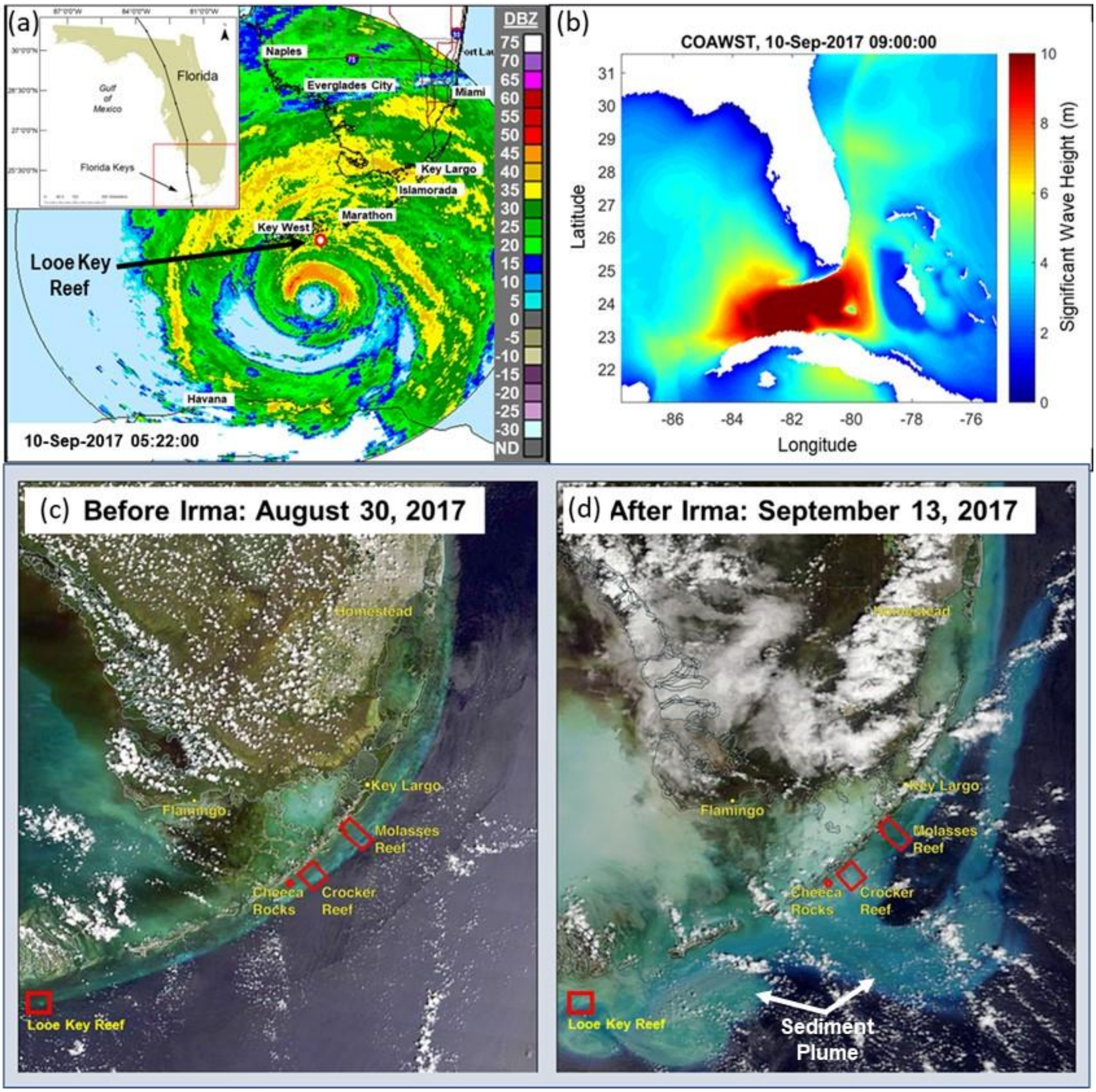

**Figure 1. Location of the Florida Keys Reef Tract, Hurricane Irma trackline and impact.** (a) NOAA, National Weather Service WSR-88D radar image (decibels, DBZ) from south Florida on 10 September 2017 at 5:22 am Eastern Daylight Time (EDT) showing approach of Hurricane Irma (inset black line = hurricane trackline). (b) Significant wave height (m) from the U.S. Geological Survey (USGS) Coupled Ocean, Atmosphere, Wave, Sediment Transport (COAWST) model on 10 September 2017 at 5:00 am EDT (Warner et al., 2010, image credit: Patricia Dalyander, USGS). (c) Satellite imagery from 30 August 2017, 11 days prior to landfall of Hurricane Irma in the Florida Keys (NASA, 2023, EOSDIS Worldview Imagery). (d) Satellite imagery from 13 September 2017, 3 days after Hurricane Irma landfall in the Florida Keys showing extensive resuspended sediment plume (NASA, 2023, EOSDIS Worldview Imagery). Red boxes show the location of Looe Key Reef relative to other reefs along the reef tract.

key reef grazer) density, size structure, and coral reef community structure before and 2.5 months after Irma at 10 locations in the Middle and Upper Florida Keys showed a significant decrease in *D. antillarum* density with increased sedimentation, suggesting sediment transport caused mortality through abrasion and burial (Kobelt et al. 2019).

While observational data from several locations indicate seafloor sediments were transported and likely caused damage to benthic habitats, the direct impact of Hurricane Irma or other tropical storms on seafloor elevation and geomorphologic structures has not previously been quantified along the FRT. In this study, we used high-resolution light-detection-and-ranging (lidar) and multibeam bathymetry data collected before and after the passage of Hurricane Irma to quantify seafloor elevation and volume change of benthic habitats and geomorphological structures resulting from the storm's impact and post-storm re-equilibration of seafloor sediments at more than 4-million point-locations at the Looe Key Reef system in the Lower FRT.

## 2 Materials and methods

### 2.1 Looe Key Reef Study Site

The FRT is the only living coral barrier reef in the continental United States, and it spans more than 580 km along the east coast of Florida from St. Lucie Inlet to the Dry Tortugas, with total reef area of approximately 1,179 km$^2$ (Finkl and Andrews, 2008; Jackson et al., 2014; Florida Department of Environmental Protection, 2022). Water depth along the FRT is up to approximately 20 m with discontinuous spur and groove formations and patch reefs separated by tidal passes, and it is characterized by both coral-dominated and non-coral dominated seafloor habitat as characterized and mapped by the Florida Fish and Wildlife Conservation Commission-Fish and Wildlife Research Institute (FWC, 2015). Much of the FRT is protected by the FKNMS, Biscayne National Park, and Dry Tortugas National Park, and includes several sanctuary preservation areas (SPAs) and special use areas (SPUs) within FKNMS, including the Looe Key SPA and SPU, that together protect over 6000 marine species (Keller and Donahue, 2006). Looe Key Reef is a barrier bank reef located approximately 10 km offshore in the Lower Florida Keys, south of Cudjoe Key, and it is characterized by a prominent, shallow reef crest with two extensive coral rubble fields, a fore reef with a spur-and-groove formation, a forereef terrace and deep reef zone, and a back reef area with seagrass communities, patch reefs, and individual coral heads (Fig. 2a-d). Seagrass beds and sandflats with intermittent patch reefs extend shoreward from Looe Key Reef proper to Hawk Channel, approximately 2 km to the north. Looe Key SPA, located at approximately 24° 32' N, 81° 24' W, is just over 18 km$^2$ and surrounds LKR proper which is less than 1.7 km$^2$. Looe Key Reef contains a coral nursery and several restoration sites for coral outplants; it is one of seven FKNMS iconic reefs, and the focus of a major collaborative habitat restoration effort known as Mission: Iconic reefs (NOAA Fisheries, 2022).

The northeastern eyewall of Hurricane Irma passed directly over LKR with the storm's center passing approximately 9 km west of LKR. However, the storm was approximately 684 km in diameter and covered the entire FRT and much of South Florida. The National Weather Service's technical summary of the storm reported tropical storm force winds more than 640

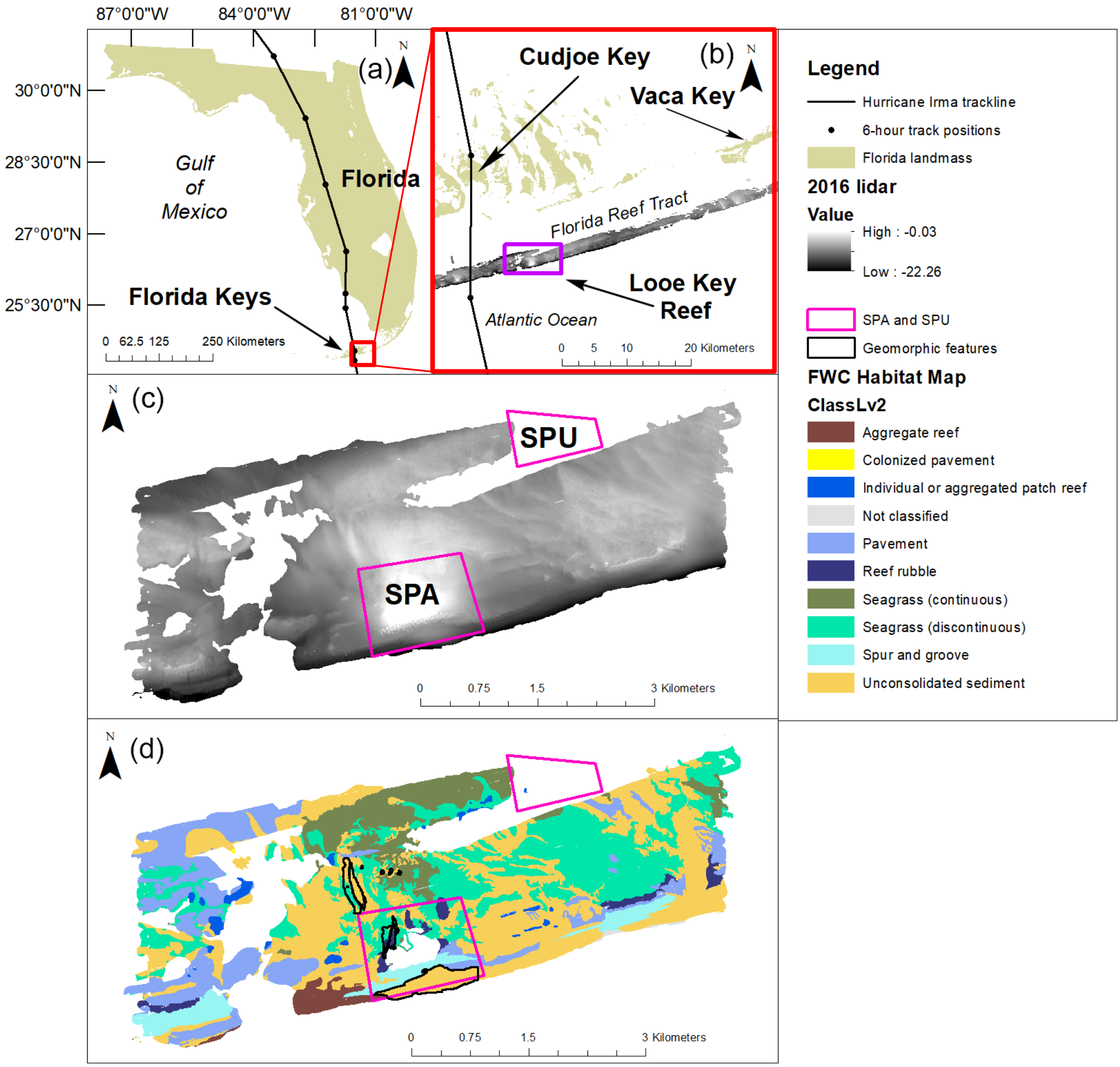

**Figure 2. Looe Key Reef location, bathymetry, and seafloor habitats.** (a) Location of the Florida Keys along the southern coast of Florida, trackline of Hurricane Irma and the location of its landfall (red box). (b) Proximity of Hurricane Irma's trackline to the Looe Key Reef study site (purple box), location of landfall at Cudjoe Key, and location of Vaca Key where the nearest NOAA-NOS (2023) Tides and Currents station was located. (c) 2016 lidar bathymetric map of the Looe Key Reef study site showing location of Florida Keys National Marine Sanctuary Special Preservation Area (SPA), Special Use Area (SPU), and geomorphic features of focused investigation for this study. (d) Habitat distribution at the Looe Key Study site from Florida Fish and Wildlife Conservation Commission-Fish and Wildlife Research Institute (2015).

km away from the storm's center, and hurricane force winds more than 125 km from the storm's center (NWS, 2022b). Gale force winds (sustained winds above 63 km $h^{-1}$) were detected by the evening of 9 September 2017 at the National Oceanic and Atmospheric Administration - National Ocean Service (NOAA-NOS), Tides and Currents Station at Vaca Key (number 8723970), 35 km to the northeast of LKR; maximum sustained winds of 213 km $h^{-1}$ were reported as the storm made landfall, and latent gale force winds were detected after the storm passed on the evening of 10 September 2017 (NOAA-NOS, 2023). The average wind direction for this period was 67.01 degrees indicating winds moved from ENE toward WSW. Wind speeds fell sharply below gale force after the storm, shifting north eastward. Wind conditions were relatively quiescent from July 2016 through January 2019 (except during Hurricane Irma) with wind speeds occasionally ranging up to approximately 56 km $h^{-1}$ during winter storms (NOAA-NOS, 2023).

## 2.2 Elevation and Habitat Data

Three Digital Elevation Models (DEMs) derived from lidar or multibeam bathymetric surveys were used for seafloor elevation- and volume-change analyses and are referenced in this study as 2016 lidar, 2017 multibeam, and 2019 lidar (Table 1).

**Table 1: Elevation datasets used in this analysis; collection dates are specific to the geographic extent of this study.**

| Digital Elevation Model | Source | Collection Dates | Horizontal Resolution/ Vertical RMSE (meters/meters) |
|---|---|---|---|
| 2016 NOAA NGS Topobathy Lidar DEM:Florida Keys Outer Reef Block 01 | Office for Coastal Management, 2017 | 23 July 2016 | 1.0/0.15 |
| Multibeam bathymetry data collected in December 2017, February and March 2018 at Looe Key, the Florida Keys | Fredericks et al, 2019 | Leg 1: 12 December 2017 – 16 December 2017<br>Leg 2: 2 February 2018 – 9 February 2018<br>Leg 3: 9 March 2018 –11 March 2018 | 1.0/0.14 |
| 2018-2019 NOAA NGS Topobathy Lidar Hurricane Irma: Miami to Marquesas Keys, FL | National Geodetic Survey, 2022 | 8 January 2019– 31 January 2019 | 1.0/0.11 |

RMSE = root mean square error.

The 2016 lidar DEM refers to data that were collected on 23 July 2016 (13.5 months before the passage of Hurricane Irma) by the NOAA Office for Coastal Management, National Geodetic Survey, Topobathy Lidar Dem Block 1 dataset (Office for Coastal Management, 2017). The 2017 multibeam DEM refers to multibeam bathymetry data collected by the U.S. Geological Survey in December 2017, and February–March 2018 at Looe Key Reef (between three and six months after the passage of Hurricane Irma), specifically to examine impacts from the storm (Fredericks et al., 2019). The 2019 lidar DEM refers to data collected January 8–31, 2019 by NOAA NGS Topobathy Lidar DEM Hurricane Irma: Miami to Marquesas Keys, FL dataset (National Geodetic Survey, 2022). The Florida Fish and Wildlife Conservation Commission (FWC) Unified Florida Reef Tract (UFRT) Map version 2.0, Level 2 habitat categories (FWC, 2015) were used to delineate geographic boundaries for 10 habitat

types within the LKR study site (Fig. 2d). Habitat labelled as 'not classified' was indistinguishable during mapping due to turbidity, cloud cover, water depth, or other interferences with obtaining an optical signature of the seafloor (Zitello et al., 2009).

### 2.3 Elevation- and Volume-Change Analyses

Seafloor elevation- and volume-change analyses were conducted using the methods of Yates et al. (2017) and 2-m grid spacing techniques of Murphy et al. (2022) (Fig. 3a). Briefly, individual geographic footprint areas (polygons) were created for each of the three 1-m resolution digital elevation models (DEMs) in ArcMap 10.7 and were used to create a common footprint polygon shapefile for the total LKR study site encompassing the overlapping area among the three datasets. The original (full areal extent, or unclipped) 2016–2017 elevation-change data set was 19.71 $km^2$ and included 4,934,364 data points. The overlapping areal extent for the 2016, 2017, and 2019 DEMs was 15.98 $km^2$ and excluded areas where water depths were too shallow for boat access to collect multibeam data in 2017 and areas of coarse interpolation within the 2017 DEM. The areal extent of each DEM was then clipped to the areal extent of the common overlapping footprint prior to elevation change analysis using the 'Clip' tool in ArcMap. The following steps were performed in Global Mapper 22.1 due to file size limitations in ArcMap. A 2-m XY grid was created in Global Mapper and clipped to the same footprint. Elevation values were then extracted from each of the three DEMs at the center points of co-aligned 2-m grid boxes. Elevation change between time periods was calculated for each of 4,007,961 paired elevation values (e.g., 2017 elevation – 2016 elevation, and 2019 elevation – 2017 elevation). Elevation-change (XYZ) point maps were generated as shapefiles for each time-period of change for the total study site; positive values indicate an increase in elevation and negative values indicate a decrease in elevation. Data are available from Fehr et al. (2021). Vertical uncertainty of elevation change analyses were estimated using methods of Yates et al. 2017 and the reported vertical accuracy of the lidar and multibeam data sets (typically reported as the 95% root-mean-square error, RMSE, Table 1) to calculate a composite RMSE ($RMSE_{Total}$) for each elevation change analysis (Fig. 3b). The RMSE of lidar and multibeam data sets used for elevation-change analyses in our study ranged from 0.11 to 0.15 m (Table 1). These values are consistent with RMSEs determined in performance evaluations of lidar sensors that ranged from 0.08 to 0.52 m (Fernandez-Diaz et al., 2014; Legleiter et al., 2016; Kinzel et al., 2013; Tonina et al., 2019; Yoshida et al., 2022). Composite RMSE values for elevation-change analyses based on comparison of lidar to multibeam DEMs ranged from 0.19 to 0.21m in our study. These values are consistent with RMSEs determined in performance evaluations of lidar sensors against multibeam echosounders that ranged from 0.02 to 0.23 m (Awadallah et al., 2023). The FWC UFRT habitat map was clipped to the intersect footprint for each elevation-change analysis using ArcMap 10.7. Each total-study-site elevation-change data set was then clipped to individual habitat polygons to create individual elevation-change shapefiles for each habitat type.

Elevation-change data from each time-period were then used to generate TIN (Triangulated Irregular Network) surface models in ArcMap for calculation of volume change. TIN models were clipped to the original overall study site intersect footprint to remove interpolation across areas where no data were collected. Lower bound (conservative) volume-change was calculated

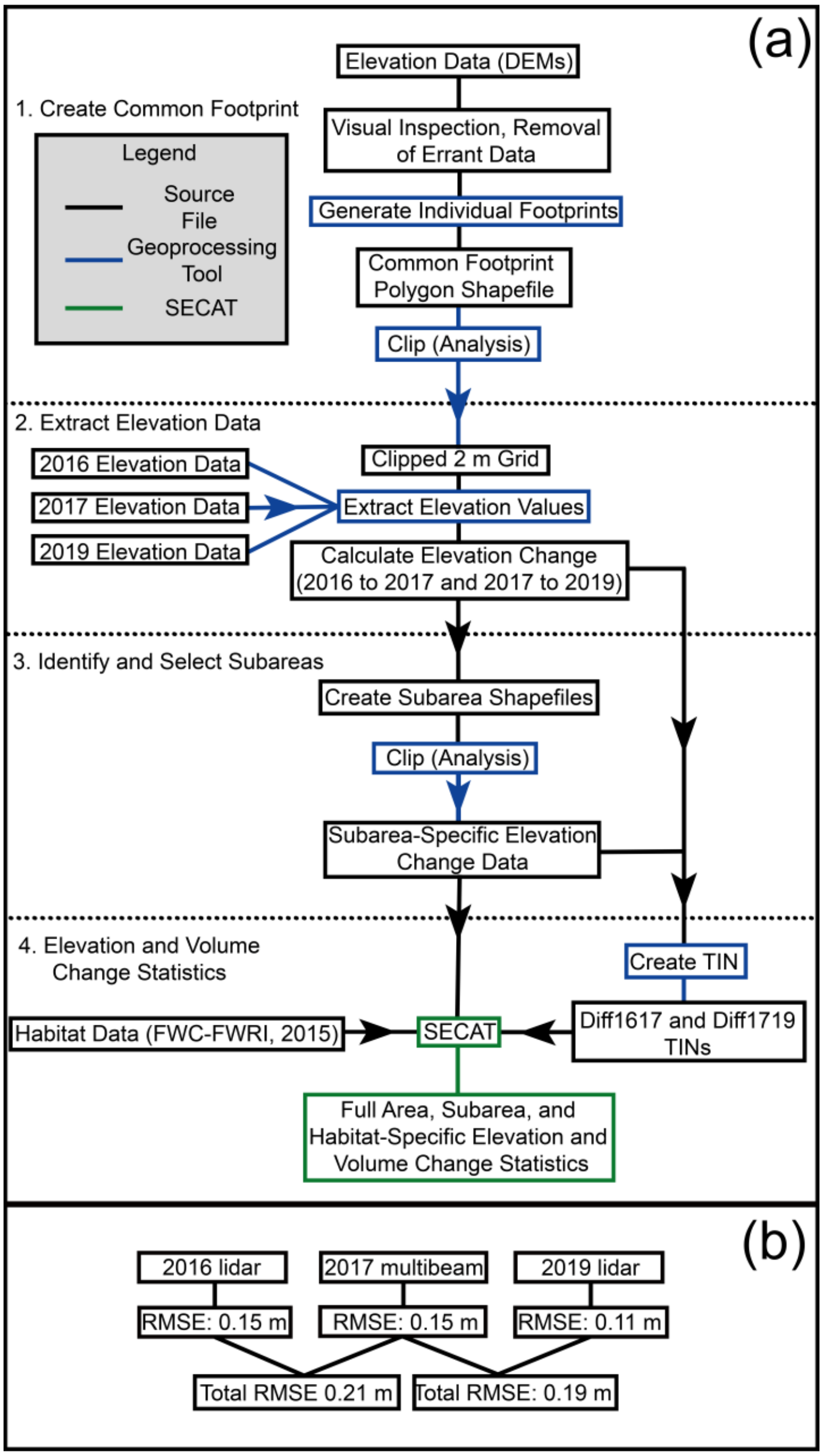

$$RMSE_{total} = \sqrt{(RMSE^2_{Source\ Dataset\ 1} + RMSE^2_{Source\ Dataset\ 2})}$$

**Figure 3. Seafloor elevation and volume change methods.** (a) Flowchart outlining generalized geoprocessing steps in ArcMap and Global Mapper (steps 1 through 3), and in the Seafloor Elevation Change Analysis Tool (SECAT, step 4) for seafloor elevation and volume change analyses based on Yates et al. (2017), Murphy et al. (2022), and Zieg and Zawada (2021). (b) Composite RMSE ($RMSE_{Total}$) for each elevation change analysis (2016 to 2017 and 2017 to 2019) calculated using reported RMSE for lidar and multibeam source data and methods of Yates et al. (2017). Black boxes indicate source data files. Blue boxes indicate steps using geoprocessing tools from ArcMap or Global Mapper. Green boxes indicate data analysis conducted using SECAT.

based on areal volume above and below surface plane heights corresponding to plus and minus the $RMSE_{Total}$ of the elevation change analysis ($RMSE_{Total}$ = 21 cm for 2016 to 2017, and 19 cm for 2017 to 2019 change analyses). Upper bound volume changes were calculated based on area volume above and below a plane height of 0 m. The attribute values stored within the elevation-change and TIN surface shapefiles were then used to compute elevation and volume change statistics for the total LKR study site and each habitat type using the Seafloor Elevation Change Analysis Tool (SECAT) custom Python script of Zieg and Zawada (2021). Pearson correlation and linear regression analyses were performed using Microsoft Excel Version 2302 (build 16.0.16130.20690) to assess relationships between mean habitat water depth, elevation change, and area-normalized volume change for each habitat type including: 1) 2016 mean water depth (estimated from mean elevation) and mean elevation change; 2) 2016 mean water depth and area-normalized volume change; 3) 2017 mean water depth and mean elevation change; 4) 2017 mean water depth and area-normalized volume change; and 5) 2017 to 2019 mean elevation change and 2016 to 2017 mean elevation.

### 2.4 Geomorphic Feature Analyses

Sub-areas or geomorphic features of high-magnitude elevation change (greater than approximately ±0.5 m) were delineated on each total-study-site elevation-change point map by manually drawing polygons in ArcMap 10.7 and creating elevation-change shapefiles for each sub-area. Each sub-area was clipped to individual habitat polygons to create individual shapefiles for each habitat type within a given sub-area. Elevation- volume-change statistics were computed for each geomorphic feature of interest, and each habitat within sub-areas of interest using SECAT and methods described in section 2.3.

We examined elevation and elevation-change along four 200 to 300 m transects across examples of high-elevation change geomorphic features. Elevation profiles for 2016, 2017, and 2019 were created for each feature of interest by extracting elevation values from each DEM along transect lines across the areas of greatest elevation change for each feature using ArcMap. Points were selected using the Select Feature by Line tool in ArcMap, and the selected features were then exported as a new shapefile. Positions and types of geomorphic features of interest were verified through in-situ observation by SCUBA divers using methods of Fehr and Yates (2020) at 30 diver reconnaissance sites throughout the total study site.

## 3 Results

### 3.1 Elevation and Volume Change Analyses

Elevation-change results for 4,007,961 point-locations at LKR between 2016–2017 (approximately 13.5 months before and 3 to 6 months after Hurricane Irma) and between 2017–2019 (from approximately 3 to 16.5 months after Hurricane Irma) are shown in Fig. 4a and b, respectively. Mean elevation-change for the total LKR study site from 2016–2017 was 0.34 m ± 0.21; and all ten habitat types (Fig. 4c) showed increases in mean elevation (accretion) ranging from 0.20 m to 0.54 m (Table 2). Largest mean elevation changes were associated with ‘aggregate reef’ (mean 2016 elevation -13.41 m) and ‘not classified’

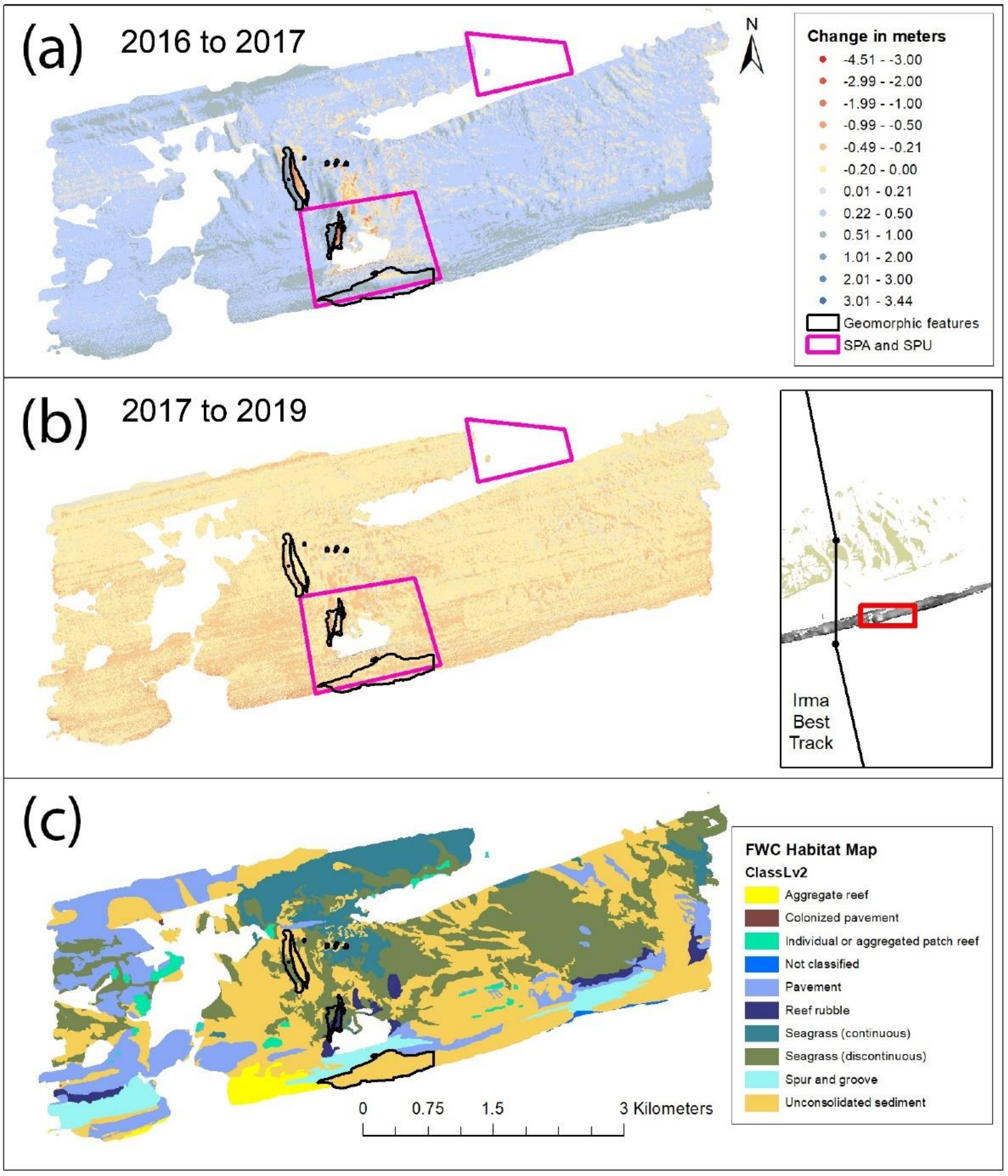

**Figure 4. Elevation-change results for 4,007,961 point-locations at Looe Key Reef.** Elevation change between (a) 2016 to 2017 (13.5 months before and 3 to 6 months after Hurricane Irma) and (b) between 2017 to 2019 (from approximately 3 to 16.5 months after Hurricane Irma), and (c) corresponding seafloor habitats (FWC, 2015). The Hurricane Irma best track data in the panel b's inset is from the NOAA NHC Irma Storm Track resource page (NHC, 2018, see also Figure 2b). Boundaries for the Looe Key Sanctuary Protection Area (SPA) and Special Protection Unit (SPU) are shown as pink polygons. Geomorphic features of interest are indicated with black polygons. Gaps in map areas indicate locations where water depth was too shallow for collection of multibeam bathymetric data.

**Table 2. Elevation change data by habitat type associated with each period and geomorphic feature subarea.**

| Habitat type | Total points (no.) | Area ($km^2$) | Mean elevation (m) 2016 | Mean elevation (m) 2017 | Mean elevation (m) 2019 | Mean elevation change (m) (SD) 2016 to 2017 | Mean elevation change (m) (SD) 2017 to 2019 |
|---|---|---|---|---|---|---|---|
| **Overall Looe Key Study Site** | | | | | | | |
| Total Study Site | 4007961 | 15.98 | -8.87 | -8.53 | -8.69 | 0.34 (0.21) | -0.15 (0.11) |
| Aggregate Reef | 76647 | 0.30 | -13.41 | -12.91 | -13.16 | 0.51 (0.20) | -0.25 (0.20) |
| Colonized Pavement | 750 | 0.0028 | -10.65 | -10.33 | -10.44 | 0.32 (0.12) | -0.11 (0.08) |
| Individual or Aggregate Patch Reef | 54414 | 0.22 | -8.66 | -8.33 | -8.51 | 0.34 (0.15) | -0.19 (0.10) |
| Not Classified | 6932 | 0.026 | -15.84 | -15.30 | -15.55 | 0.54 (0.25) | -0.25 (0.17) |
| Pavement | 645001 | 2.57 | -10.00 | -9.62 | -9.79 | 0.37 (0.16) | -0.16 (0.11) |
| Reef Rubble | 80987 | 0.32 | -6.19 | -5.99 | -6.17 | 0.20 (0.36) | -0.18 (0.12) |
| Seagrass Continuous | 402458 | 1.60 | -7.69 | -7.42 | -7.54 | 0.27 (0.18) | -0.12 (0.09) |
| Seagrass Discontinuous | 1067504 | 4.26 | -7.24 | -6.96 | -7.10 | 0.28 (0.21) | -0.14 (0.10) |
| Spur and Groove | 184875 | 0.74 | -9.82 | -9.45 | -9.65 | 0.37 (0.25) | -0.19 (0.19) |
| Unconsolidated Sediment | 1488416 | 5.94 | -9.63 | -9.26 | -9.42 | 0.37 (0.21) | -0.16 (0.09) |
| **Sand Wave** | | | | | | | |
| Total Accretion Area | 15336 | 0.060 | -6.32 | -5.53 | -5.68 | 0.79 (0.45) | -0.15 (0.12) |
| Seagrass Discontinuous | 7345 | 0.029 | -5.98 | -5.08 | -5.23 | 0.90 (0.49) | -0.15 (0.13) |
| Unconsolidated Sediment | 7991 | 0.031 | -6.63 | -5.95 | -6.09 | 0.68 (0.37) | -0.14 (0.10) |
| Total Erosion Area | 11265 | 0.043 | -5.40 | -5.75 | -5.90 | -0.36 (0.28) | -0.15 (0.06) |
| Seagrass Discontinuous | 580 | 0.002 | -5.72 | -5.87 | -6.02 | -0.15 (0.15) | -0.15 (0.08) |
| Unconsolidated Sediment | 10685 | 0.041 | -5.38 | -5.74 | -5.90 | -0.37 (0.29) | -0.15 (0.05) |
| **Scour Marks** | | | | | | | |
| Scour Mark 1 | 202 | 0.00071 | -7.03 | -7.51 | -7.41 | -0.49 (0.26) | 0.10 (0.12) |
| Seagrass Discontinuous | 197 | 0.00071 | -7.02 | -7.51 | -7.41 | -0.49 (0.26) | 0.11 (0.12) |
| Unconsolidated Sediment | 5 | <0.00001 | -7.34 | -7.47 | -7.52 | -0.12 (0.03) | -0.05 (0.02) |
| Scour Mark 2 | 388 | 0.0014 | -5.41 | -5.91 | -5.71 | -0.50 (0.27) | 0.20 (0.20) |
| Seagrass Continuous | 338 | 0.00124 | -5.41 | -5.94 | -5.70 | -0.53 (0.27) | 0.24 (0.18) |
| Unconsolidated Sediment | 50 | 0.00016 | -5.42 | -5.67 | -5.72 | -0.26 (0.16) | -0.05 (0.13) |
| Scour Mark 3 | 518 | 0.00188 | -5.64 | -6.14 | -6.02 | -0.50 (0.29) | 0.12 (0.19) |
| Seagrass Continuous | 518 | 0.00188 | -5.64 | -6.14 | -6.02 | -0.50 (0.29) | 0.12 (0.19) |
| Scour Mark 4 | 417 | 0.00152 | -5.20 | -5.74 | -5.63 | -0.54 (0.28) | 0.12 (0.21) |
| Seagrass Continuous | 411 | 0.00151 | -5.19 | -5.74 | -5.62 | -0.55 (0.27) | 0.12 (0.21) |
| Unconsolidated Sediment | 6 | 0.00001 | -5.69 | -5.74 | -5.84 | -0.06 (0.05) | -0.10 (0.05) |
| **Reef Rubble Field** | | | | | | | |
| Total Accretion Area | 7216 | 0.028 | -4.22 | -3.32 | -3.57 | 0.89 (0.45) | -0.24 (0.30) |
| Reef Rubble | 3102 | 0.012 | -3.71 | -2.84 | -3.05 | 0.87 (0.44) | -0.21 (0.36) |
| Seagrass Discontinuous | 3489 | 0.014 | -4.66 | -3.67 | -3.97 | 0.99 (0.42) | -0.30 (0.24) |
| Unconsolidated Sediment | 628 | 0.00237 | -4.25 | -3.82 | -3.91 | 0.43 (0.26) | -0.10 (0.12) |
| Total Erosion Area | 6043 | 0.023 | -3.00 | -3.64 | -3.74 | -0.63 (0.48) | -0.10 (0.19) |
| Reef Rubble | 3409 | 0.013 | -2.61 | -3.39 | -3.44 | -0.77 (0.50) | -0.06 (0.20) |
| Seagrass Discontinuous | 1941 | 0.00708 | -3.51 | -4.05 | -4.22 | -0.54 (0.43) | -0.17 (0.15) |
| Unconsolidated Sediment | 694 | 0.00248 | -3.50 | -3.70 | -3.82 | -0.20 (0.15) | -0.12 (0.13) |
| **Sand Lobe** | | | | | | | |
| Total Area | 67389 | 0.266 | -12.41 | -11.90 | -12.10 | 0.51 (0.29) | -0.20 (0.09) |
| Unconsolidated Sediment | 67389 | 0.266 | -12.41 | -11.90 | -12.10 | 0.51 (0.29) | -0.20 (0.09) |

*58 data points fell on borders between habitats and were counted twice during habitat analysis. SD = standard deviation.

(mean 2016 elevation -15.84 m) habitat types. Smallest mean elevation changes were associated with 'reef rubble' (mean 2016 elevation -6.19 m) and 'seagrass continuous' (mean 2016 elevation -7.69 m) habitats (Table 2). Only 4% of all data points showed losses in elevation (erosion) ranging from -0.01 m to -0.44 m, while 96% of all data points showed gains in elevation ranging from 0.31 m to 0.55 m across all habitats. Pearson correlation analysis showed a very strong positive correlation ($r(8) = 0.96$, $p = 0.000$) and linear relationship ($r^2 = 0.92$, Fig. 5a) between 2016 mean habitat water depth (estimated from mean elevation) and mean elevation change; mean elevation gains increased significantly with increasing water depth (i.e., decreasing seafloor elevation). Net volume change was up to 5.36 $mM^3$ over the total 15.98 $km^2$ Looe Key study site; and all

habitat types showed increases in net volume (accretion) with upper bound ranges from 0.001 to 2.19 $Mm^3$ (Table 3). Largest net volume changes were associated with habitats covering the largest areal extent of the study area including 'pavement,' 'discontinuous seagrass,' and 'unconsolidated sediment.' Pearson correlation analysis also indicated a very strong positive correlation (r(8) = 0.99, p = 0.000) and linear relationship ($r^2$ = 0.92, Fig. 5b)) between 2016 mean habitat water depth and area-normalized volume change; area-normalized volume gains increased significantly with increasing water depth. Largest area-normalized volume changes of 0.51 $mM^3$ and 0.54 $mM^3$ were observed for 'aggregate reef' and 'not classified' habitats, respectively; and smallest changes of 0.20 to 0.27 $mM^3$ were observed for 'reef rubble' and 'continuous seagrass' habitats (Table 3), consistent with mean elevation changes for those habitats. Mean elevation-change values of the 2016–2017 elevation change data set that was clipped to an area of 15.98 $km^2$ and used for this analysis were within ±0.01 m, and area-normalized volumes were within ±0.016 $Mm^3$ $km^{-2}$, of values calculated in the original 19.71 $km^2$ published data set (unclipped) for the overall study site and all habitats (Yates et al., 2019).

Mean elevation-change during a 13-month time-period between December 2017 to June 2019 (up to approximately 16.5 months after Hurricane Irma) was -0.15 ± 0.11 m, and all habitat types showed losses in mean elevation ranging from -0.11 m to -0.25 m (Fig. 4b, Table 2). Largest mean elevation changes were associated with 'aggregate reef' and 'not classified' habitat types, and smallest changes were associated with 'colonized pavement' and 'continuous seagrass' habitats (Table 2). Only 5% of all data points showed gains in elevation with mean accretion ranging from 0.04 m to 0.19 m, while 95% of all data points showed losses in elevation with mean erosion ranging from -0.13 m to -0.27 m across all habitat types. Pearson correlation analysis indicated a moderate correlation (r(8) = -0.67, p = 0.035) and linear relationship ($r^2$ = 0.45, Fig. 5c) between estimated 2017 mean habitat water depth and mean elevation change; mean elevation loss generally increased with increasing water depth. Net volume change was up to -2.46 $mM^3$ over the total 15.98 $km^2$ Looe Key study site and area-normalized volume change was -0.15 $Mm^3km^{-2}$. Losses in net volume up to -0.931 $Mm^3$ (erosion) were observed across all habitat types (Table 4).

Largest net volume changes were associated with habitats covering the largest areal extent of the study area including 'pavement,' 'discontinuous seagrass,' and 'unconsolidated sediment.' Pearson correlation analysis indicated a moderate correlation (r(8) = -0.67, p = 0.035) and linear relationship ($r^2$ = 0.45, Fig. 5d) between 2017 mean habitat water depth and area-normalized volume change; area-normalized volume losses generally increased with increasing water depth. Largest area-normalized volume changes were observed for 'aggregate reef' and 'not classified' habitats, -0.254 and -0.247 $Mm^3$ $km^{-2}$, respectively; smallest changes were observed for 'colonized pavement' and 'continuous seagrass' habitats, -0.112 to -0.118 $Mm^3$ $km^{-2}$ respectively (Table 4), consistent with mean elevation changes for those habitats. Pearson correlation analysis indicated a strong negative correlation (r(8) = -0.74, p = 0.014) and linear relationship ($r^2$ = 0.55, Fig. 5e) between 2017 to 2019 mean habitat elevation change and 2016 to 2017 mean habitat elevation change; mean elevation losses during 2017 to 2019 were significantly greater in habitats with larger mean elevation gains during 2016 to 2017. Mean elevation change (loss)

during 2017 to 2019 was 35 to 55% of the mean elevation change (gain) during 2016 to 2017 for all habitats except for reef rubble which was 92% and had the shallowest mean depth (6.0 m) of all habitats.

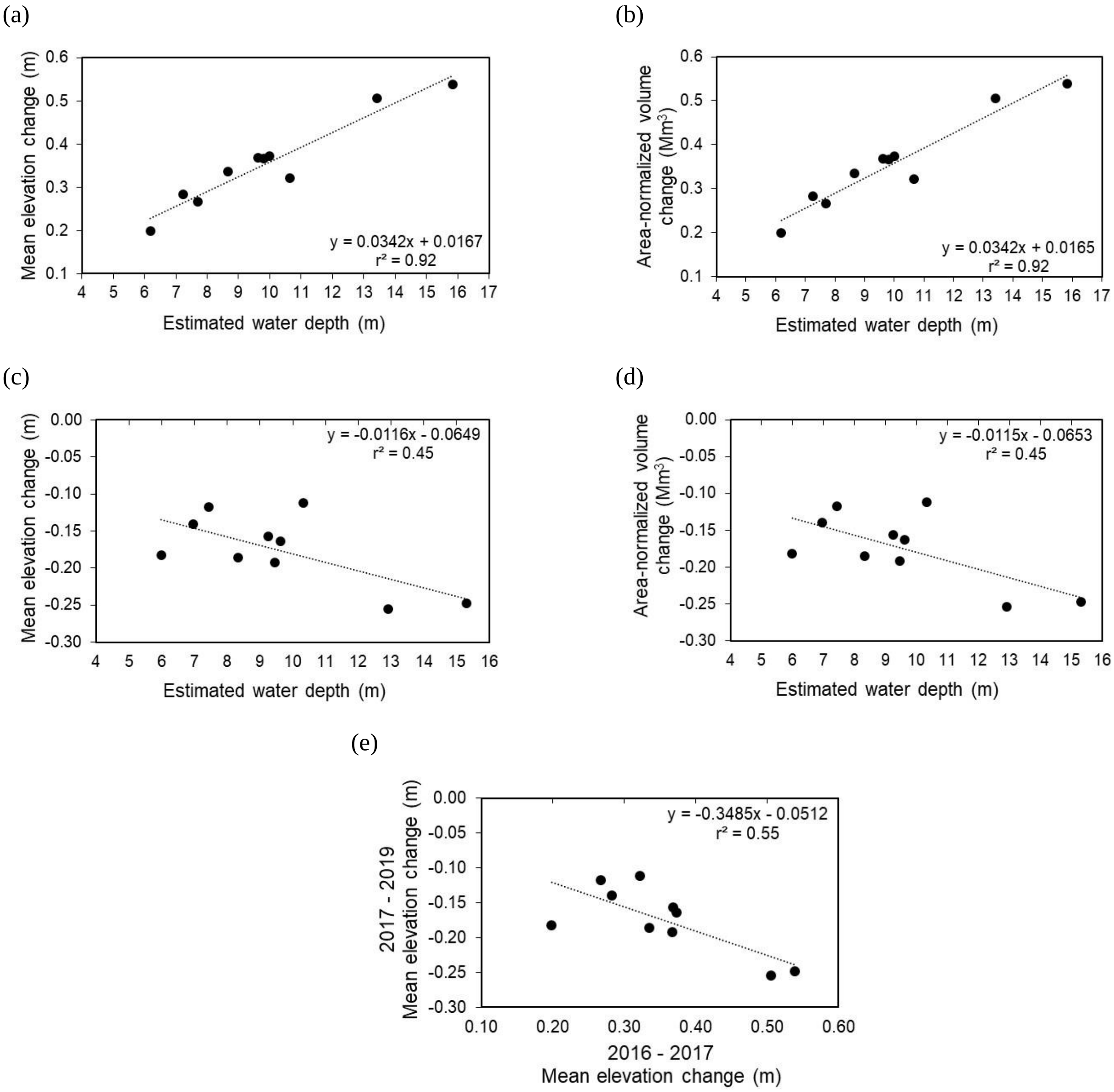

**Figure 5. Linear relationships between elevation change, volume change, and water depth.** Linear relationships and coefficients of determination between (a) mean elevation change, (b) mean area-normalized volume change, and estimated 2016 mean water depth for seafloor habitats of the Looe Key study site between 2016 to 2017. Linear relationships and coefficients of determination between (c) mean elevation change, (d) mean area-normalized volume change, and estimated 2017 mean water depth for seafloor habitats of the Looe Key study site between 2017 to 2019 (a, b). Linear relationship between 2017 to 2019 mean elevation change and 2016 to 2017 mean elevation change (e).

**Table 3. Compiled volume change data by habitat type for all study areas during the 2016 to 2017 study period (storm period).**

| Habitat Type | Habitat Area ($km^2$) | Gross erosion ($Mm^3$) | | Gross accretion ($Mm^3$) | | Net volume change ($Mm^3$ study-area$^{-1}$) | | Area-normalized volume change ($Mm^3$ $km^{-2}$) | |
|---|---|---|---|---|---|---|---|---|---|
| | | Lower | Upper | Lower | Upper | Lower | Upper | Lower | Upper |
| **Overall Looe Key Study Site** | | | | | | | | | |
| Total Study Site | 15.98 | 0.053 | 0.134 | 2.456 | 5.490 | 2.403 | 5.356 | 0.150 | 0.335 |
| Aggregate Reef | 0.30 | 0 | 0 | 0.090 | 0.154 | 0.090 | 0.154 | 0.296 | 0.505 |
| Colonized Pavement | 0.0028 | 0 | 0 | 0 | 0.001 | 0 | 0.001 | 0.124 | 0.323 |
| Individual or Aggregate Patch Reef | 0.22 | 0 | 0 | 0.029 | 0.073 | 0.029 | 0.072 | 0.136 | 0.336 |
| Not Classified | 0.026 | 0 | 0 | 0.009 | 0.014 | 0.009 | 0.014 | 0.337 | 0.540 |
| Pavement | 2.57 | 0 | 0.003 | 0.445 | 0.962 | 0.445 | 0.959 | 0.173 | 0.373 |
| Reef Rubble | 0.32 | 0.013 | 0.021 | 0.033 | 0.085 | 0.020 | 0.064 | 0.062 | 0.199 |
| Seagrass Continuous | 1.60 | 0.008 | 0.021 | 0.152 | 0.449 | 0.144 | 0.428 | 0.090 | 0.267 |
| Seagrass Discontinuous | 4.26 | 0.015 | 0.045 | 0.477 | 1.250 | 0.462 | 1.205 | 0.109 | 0.283 |
| Spur and Groove | 0.74 | 0.003 | 0.007 | 0.136 | 0.278 | 0.133 | 0.271 | 0.181 | 0.367 |
| Unconsolidated Sediment | 5.94 | 0.013 | 0.037 | 1.083 | 2.226 | 1.070 | 2.189 | 0.180 | 0.369 |
| **Sand Wave** | | | | | | | | | |
| Total Accretion Area | 0.060 | 0 | 0 | 0.036 | 0.048 | 0.036 | 0.048 | 0.598 | 0.800 |
| Seagrass Discontinuous | 0.029 | 0 | 0 | 0.020 | 0.026 | 0.020 | 0.026 | 0.709 | 0.914 |
| Unconsolidated Sediment | 0.031 | 0 | 0 | 0.015 | 0.021 | 0.015 | 0.021 | 0.494 | 0.694 |
| Total Erosion Area | 0.043 | 0.009 | 0.016 | 0 | 0 | -0.009 | -0.016 | -0.198 | -0.370 |
| Seagrass Discontinuous | 0.002 | <0.001 | <0.001 | 0 | 0 | <-0.001 | <-0.001 | -0.045 | -0.162 |
| Unconsolidated Sediment | 0.041 | 0.008 | 0.016 | 0 | 0 | -0.008 | -0.016 | -0.205 | -0.380 |
| **Scour Marks** | | | | | | | | | |
| Scour Mark 1 | 0.00071 | 0.0002 | 0.0004 | 0 | 0 | -0.0002 | -0.0004 | -0.3083 | -0.5114 |
| Seagrass Discontinuous | 0.00071 | 0.0002 | 0.0004 | 0 | 0 | -0.0002 | -0.0004 | -0.3118 | -0.5154 |
| Unconsolidated Sediment | <0.00001 | 0 | 0 | 0 | 0 | 0 | <-0.0001 | -0.0001 | -0.1479 |
| Scour Mark 2 | 0.0014 | 0.0005 | 0.0007 | 0 | 0 | -0.0005 | -0.0007 | -0.3255 | -0.5271 |
| Seagrass Continuous | 0.00124 | 0.0004 | 0.0007 | 0 | 0 | -0.0004 | -0.0007 | -0.3558 | -0.5595 |
| Unconsolidated Sediment | 0.00016 | 0 | <-0.0001 | 0 | 0 | 0 | <-0.0001 | -0.0943 | -0.2790 |
| Scour Mark 3 | 0.00188 | 0.0006 | 0.0010 | 0 | 0 | -0.0006 | -0.0010 | -0.3247 | -0.5232 |
| Seagrass Continuous | 0.00188 | 0.0006 | 0.0010 | 0 | 0 | -0.0006 | -0.0010 | -0.3247 | -0.5232 |
| Scour Mark 4 | 0.00152 | 0.0006 | 0.0009 | 0 | 0 | -0.0006 | -0.0009 | -0.3631 | -0.5661 |
| Seagrass Continuous | 0.00151 | 0.0006 | 0.0009 | 0 | 0 | -0.0006 | -0.0009 | -0.3658 | -0.5697 |
| Unconsolidated Sediment | 0.00001 | 0 | <-0.0001 | 0 | 0 | 0 | <-0.0001 | 0.0000 | -0.0748 |
| **Reef Rubble Field** | | | | | | | | | |
| Total Accretion Area | 0.028 | 0 | 0 | 0.020 | 0.025 | 0.020 | 0.025 | 0.707 | 0.914 |
| Reef Rubble | 0.012 | 0 | 0 | 0.008 | 0.011 | 0.008 | 0.011 | 0.690 | 0.897 |
| Seagrass Discontinuous | 0.014 | 0 | 0 | 0.011 | 0.014 | 0.011 | 0.014 | 0.802 | 1.011 |
| Unconsolidated Sediment | 0.002 | 0 | 0 | 0.001 | 0.001 | 0.001 | 0.001 | 0.252 | 0.446 |
| Total Erosion Area | 0.023 | 0.011 | 0.015 | 0 | 0 | -0.011 | -0.015 | -0.464 | -0.661 |
| Reef Rubble | 0.013 | 0.008 | 0.010 | 0 | 0 | -0.008 | -0.010 | -0.584 | -0.788 |
| Seagrass Discontinuous | 0.007 | 0.003 | 0.004 | 0 | 0 | -0.003 | -0.004 | -0.382 | -0.577 |
| Unconsolidated Sediment | 0.002 | 0.000 | 0.001 | 0 | 0 | 0.000 | -0.001 | -0.064 | -0.221 |
| **Sand Lobe** | | | | | | | | | |
| Total Area | 0.27 | 0 | 0.002 | 0.089 | 0.139 | 0.089 | 0.137 | 0.332 | 0.513 |
| Unconsolidated Sediment | 0.27 | 0 | 0.002 | 0.089 | 0.139 | 0.089 | 0.137 | 0.332 | 0.513 |

'Upper' and 'lower' headings refer to the upper and lower bounds of volume change based on total RMSE root mean square error). Lower bounds use total RMSE as a plane height in calculating volume.

**Table 4. Compiled volume change data by habitat type for all study areas during the 2017 to 2019 study period (post-storm re-equilibration period).**

| Habitat Type | Habitat Area ($km^2$) | Gross erosion ($Mm^3$) | | Gross accretion ($Mm^3$) | | Net volume change ($Mm^3$ study-area$^{-1}$) | | Area-normalized volume change ($Mm^3\ km^{-2}$) | |
|---|---|---|---|---|---|---|---|---|---|
| | | Lower | Upper | Lower | Upper | Lower | Upper | Lower | Upper |
| **Overall Looe Key Study Site** | | | | | | | | | |
| Total Study Site | 15.98 | 0.316 | 2.502 | 0.005 | 0.041 | -0.311 | -2.461 | -0.019 | -0.154 |
| Aggregate Reef | 0.30 | 0.028 | 0.078 | 0 | <0.001 | -0.028 | -0.077 | -0.093 | -0.254 |
| Colonized Pavement | 0.0028 | <0.001 | <0.001 | 0 | 0 | <-0.001 | <-0.001 | -0.004 | -0.112 |
| Individual or Aggregate Patch Reef | 0.22 | 0.006 | 0.040 | 0 | 0 | -0.006 | -0.040 | -0.028 | -0.186 |
| Not Classified | 0.026 | 0.002 | 0.007 | 0 | <0.001 | -0.002 | -0.006 | -0.083 | -0.247 |
| Pavement | 2.57 | 0.059 | 0.424 | 0 | 0.004 | -0.059 | -0.420 | -0.023 | -0.163 |
| Reef Rubble | 0.32 | 0.010 | 0.061 | 0.001 | 0.002 | -0.009 | -0.059 | -0.029 | -0.182 |
| Seagrass Continuous | 1.60 | 0.012 | 0.197 | 0.001 | 0.008 | -0.011 | -0.189 | -0.007 | -0.118 |
| Seagrass Discontinuous | 4.26 | 0.064 | 0.612 | 0.001 | 0.015 | -0.063 | -0.597 | -0.015 | -0.140 |
| Spur and Groove | 0.74 | 0.032 | 0.145 | 0.001 | 0.003 | -0.031 | -0.141 | -0.042 | -0.192 |
| Unconsolidated Sediment | 5.94 | 0.102 | 0.938 | 0.000 | 0.007 | -0.102 | -0.931 | -0.017 | -0.157 |
| **Sand Wave** | | | | | | | | | |
| Total Accretion Area | 0.060 | 0.0015 | 0.0093 | 0 | 0.0005 | -0.0015 | -0.0088 | -0.0245 | -0.1479 |
| Seagrass Discontinuous | 0.029 | 0.0010 | 0.0048 | 0 | 0.0003 | -0.0010 | -0.0044 | -0.0336 | -0.1544 |
| Unconsolidated Sediment | 0.031 | 0.0005 | 0.0045 | 0 | 0.0001 | -0.0005 | -0.0044 | -0.0159 | -0.1419 |
| Total Erosion Area | 0.043 | 0.0003 | 0.0066 | 0 | 0 | -0.0003 | -0.0066 | -0.0074 | -0.1529 |
| Seagrass Discontinuous | 0.002 | <0.0001 | 0.0003 | 0 | 0 | <-0.0001 | -0.0003 | -0.0158 | -0.1521 |
| Unconsolidated Sediment | 0.041 | 0.0003 | 0.0063 | 0 | 0 | -0.0003 | -0.0063 | -0.0070 | -0.1529 |
| **Scour Marks** | | | | | | | | | |
| Scour Mark 1 | 0.00071 | 0.0000 | 0.0000 | <0.0001 | 0.0001 | <0.0001 | 0.0001 | 0.0171 | 0.1201 |
| Seagrass Discontinuous | 0.00071 | 0.0000 | 0.0000 | <0.0001 | 0.0001 | <0.0001 | 0.0001 | 0.0173 | 0.1219 |
| Unconsolidated Sediment | <0.00001 | 0.0000 | <0.0001 | 0.0000 | 0.0000 | 0.0000 | <-0.0001 | 0.0000 | -0.0447 |
| Scour Mark 2 | 0.0014 | 0 | 0 | 0.0001 | 0.0003 | 0.0001 | 0.0003 | 0.0880 | 0.2226 |
| Seagrass Continuous | 0.00124 | 0 | 0 | 0.0001 | 0.0003 | 0.0001 | 0.0003 | 0.0996 | 0.2550 |
| Unconsolidated Sediment | 0.00016 | <0.0001 | <0.0001 | 0 | 0 | <-0.0001 | <-0.0001 | -0.0009 | -0.0254 |
| Scour Mark 3 | 0.00188 | 0 | 0 | 0.0001 | 0.0003 | 0.0001 | 0.0003 | 0.0524 | 0.1380 |
| Seagrass Continuous | 0.00188 | 0 | 0 | 0.0001 | 0.0003 | 0.0001 | 0.0003 | 0.0524 | 0.1380 |
| Scour Mark 4 | 0.00152 | 0 | 0 | 0.0001 | 0.0002 | 0.0001 | 0.0002 | 0.0615 | 0.1334 |
| Seagrass Continuous | 0.00151 | 0 | 0 | 0.0001 | 0.0002 | 0.0001 | 0.0002 | 0.0620 | 0.1351 |
| Unconsolidated Sediment | 0.00001 | 0.0000 | <0.0001 | 0.0000 | 0.0000 | 0.0000 | <-0.0001 | 0.0000 | -0.1029 |
| **Reef Rubble Field** | | | | | | | | | |
| Total Accretion Area | 0.028 | 0.0040 | 0.0080 | 0.0010 | 0.0010 | -0.0040 | -0.0070 | -0.1310 | -0.2480 |
| Reef Rubble | 0.012 | 0.0020 | 0.0040 | 0 | 0.0010 | -0.0010 | -0.0030 | -0.1260 | -0.2180 |
| Seagrass Discontinuous | 0.014 | 0.0020 | 0.0040 | 0 | 0 | -0.0020 | -0.0040 | -0.1560 | -0.3020 |
| Unconsolidated Sediment | 0.002 | <0.0001 | <0.0001 | 0 | 0 | <-0.0001 | <-0.0001 | -0.0070 | -0.0910 |
| Total Erosion Area | 0.023 | 0.0005 | 0.0031 | 0.0003 | 0.0008 | -0.0002 | -0.0023 | -0.0084 | -0.1026 |
| Reef Rubble | 0.013 | 0.0001 | 0.0015 | 0.0003 | 0.0007 | 0.0002 | -0.0008 | 0.0113 | -0.0612 |
| Seagrass Discontinuous | 0.007 | 0.0003 | 0.0013 | 0 | 0.0001 | -0.0003 | -0.0012 | -0.0429 | -0.1741 |
| Unconsolidated Sediment | 0.002 | <0.0001 | 0.0003 | 0 | 0 | <-0.0001 | -0.0003 | -0.0152 | -0.1188 |
| **Sand Lobe** | | | | | | | | | |
| Total Area | 0.27 | 0.010 | 0.055 | 0 | <0.001 | -0.010 | -0.054 | -0.038 | -0.204 |
| Unconsolidated Sediment | 0.27 | 0.010 | 0.055 | 0 | <0.001 | -0.010 | -0.054 | -0.038 | -0.204 |

'Upper' and 'lower' headings refer to the upper and lower bounds of volume change based on total RMSE (root mean square error).

### 3.2 Geomorphic Feature Analyses

Large-scale geomorphic features that were 10s to 100s of $m^2$ in areal extent and showed extensive erosion and/or accretion with elevation-changes greater than 0.5 m were observed between 2016 and 2017 (Fig. 6 and 7). Examples of these features included migration of a sand wave in the back reef area of Looe Key reef indicated by adjacent areas of erosion and accretion.

Several scour features developed in 'discontinuous seagrass' and 'unconsolidated sediment' habitats of the Looe Key back reef, indicated by areas of erosion that appear as pits. Rubble fields within and near the Looe Key SPA area were displaced, as indicated by adjacent areas of accretion and erosion. Substantial deposition of sediments occurred along a sand lobe at the base of the Looe Key Reef 'spur and groove' habitat.

### 3.2.1 Sand Wave

Migration of a sand wave was observed in the back reef area of Looe Key Reef between 2016 and 2017, with minor erosion of this feature occurring between 2017 and 2019 (Fig. 6a, b, and c). The sand wave was approximately 733 m long and 104 m wide at its widest point in 2017, 2 m in height from the crest to base on the deepest (western) edge, with average water depth of approximately 5.6 m. Transect elevation profiles showed the location of this feature in 2016, westward migration of approximately 78 m (crest to crest) in 2017, and minor erosion in 2019 (Fig. 7a). An accretion of 0.060 $km^2$ included approximately 50% discontinuous seagrass and 50% unconsolidated sediment habitat.

Between 2016 and 2017, mean elevation change of the accretion area (2017 location of the sand wave) was 0.79 m (Table 2) with a maximum elevation gain at the crest of 1.84m. An adjacent area of erosion was approximately 630 x 122 m in length and width (0.043 $km^2$) and included approximately 5% discontinuous seagrass and 95% unconsolidated sediment. Mean elevation-change of the erosion area was -0.36 m (Table 2) with a maximum elevation loss of -1.23 m near the 2016 location of the sand wave crest. Total net volume change for the accretion area of the feature was 0.048 $Mm^3$ and area-normalized volume change was 0.800 $Mm^3$ $km^{-2}$ (Table 3). Mean elevation-change and area-normalized volume change was greatest within the discontinuous seagrass habitat (0.90 m and 0.914 $Mm^3$ $km^{-2}$, respectively), approximately 2.7 times greater than mean elevation change and area-normalized volume change for the overall Looe Key study site. It accounted for 55% of total net volume gain, indicating burial of seagrass habitat during migration of the sand wave. Net volume change of the erosion area was approximately -0.016 $Mm^3$ and area-normalized volume change -0.37 $Mm^3$ $km^{-2}$ with 98% of net volume change associated with erosion of unconsolidated sediment habitat (Table 3).

Between 2017 and 2019, the sand wave (accretion area) showed mean elevation and net volume change of approximately -0.15 m and approximately -0.009 $Mm^3$, respectively (Table 2, 4 and Fig. 6b). Similar mean elevation change values were observed for discontinuous seagrass and unconsolidated sediment habitats associated with the feature, and net volume change for each habitat was approximately 50% of the total net volume change (Table 4). Area-normalized volume change was similar for the total area of the sand wave and the sub-areas within it, including discontinuous seagrass and unconsolidated sediment habitats, ranging from approximately -0.148 to -0.154 $Mm^3/km^2$. The adjacent erosion area (original 2016 location of the sand wave) also showed a mean elevation change of -0.15 m with similar values for the associated discontinuous seagrass and unconsolidated sediment habitats. Net volume change of the erosion area was approximately -0.007 $Mm^3$ with approximately

95% of this loss associated with unconsolidated sediment (Table 4). Area-normalized volume change was also consistent across the total erosion feature area, discontinuous seagrass, and unconsolidated sediment habitats at -0.15 $Mm^3$ $km^{-2}$.

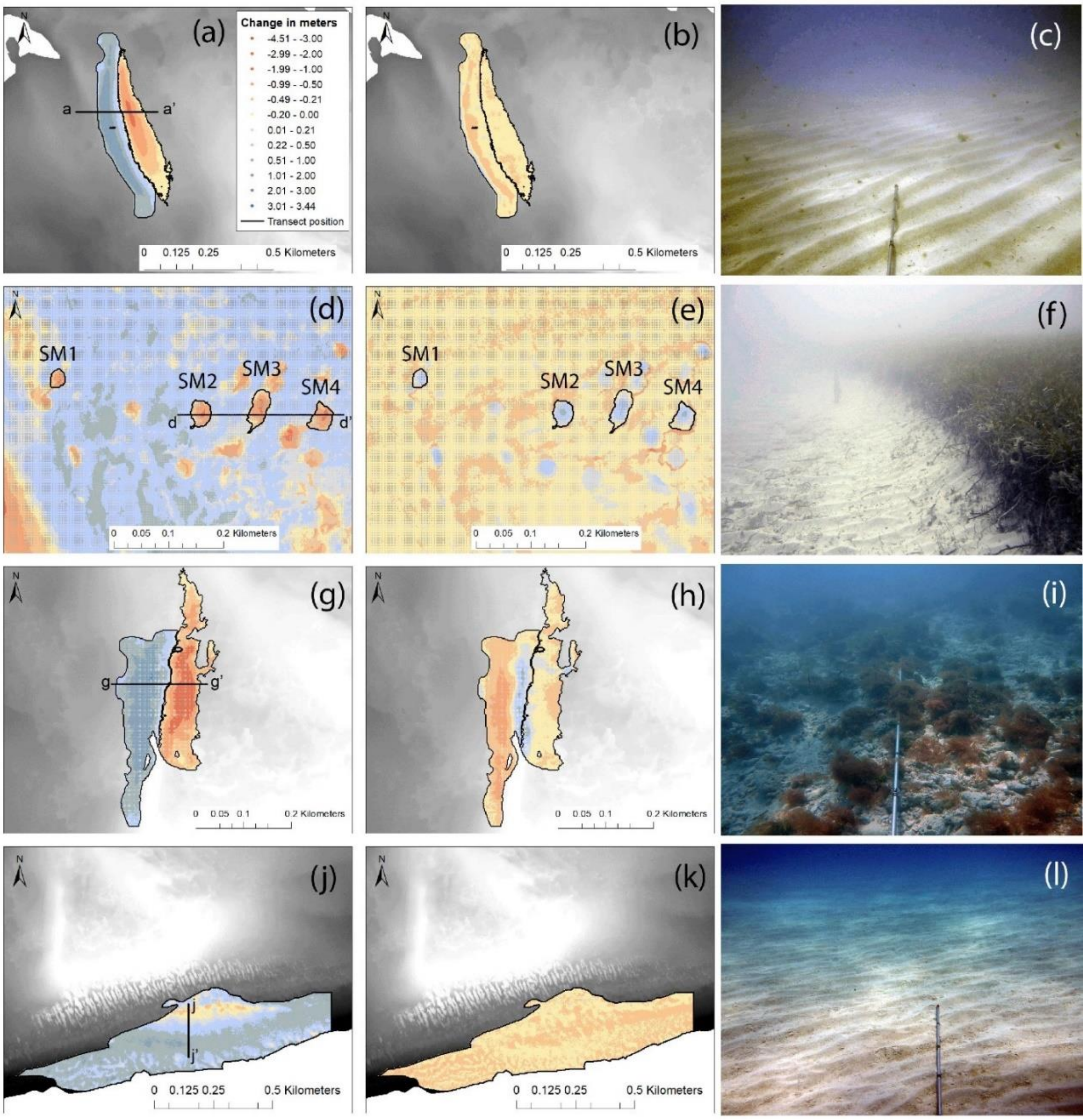

**Figure 6. Elevation change data and transect positions for each geomorphic feature subarea.** Geomorphic features included a sand wave (a, b, c), scour marks (d, e, f), western rubble field (g, h, i), and sand lobe subareas (j, k, l). These feature locations and corresponding habitat are also shown in Fig. 4. Elevation-change from 2016 to 2017 (a, d, g, j), 2017 to 2019 (b, e, h, k), and corresponding reconnaissance imagery (c, f, i, l) Transect positions are indicated by black lines and lowercase letters in the elevation change panels (see also Fig. 7). Scour marks in panels d and e are labelled SM1 through 4. Photo credit: Mitch Lemon, Cherokee Nations System Solutions for U.S. Geological Survey.

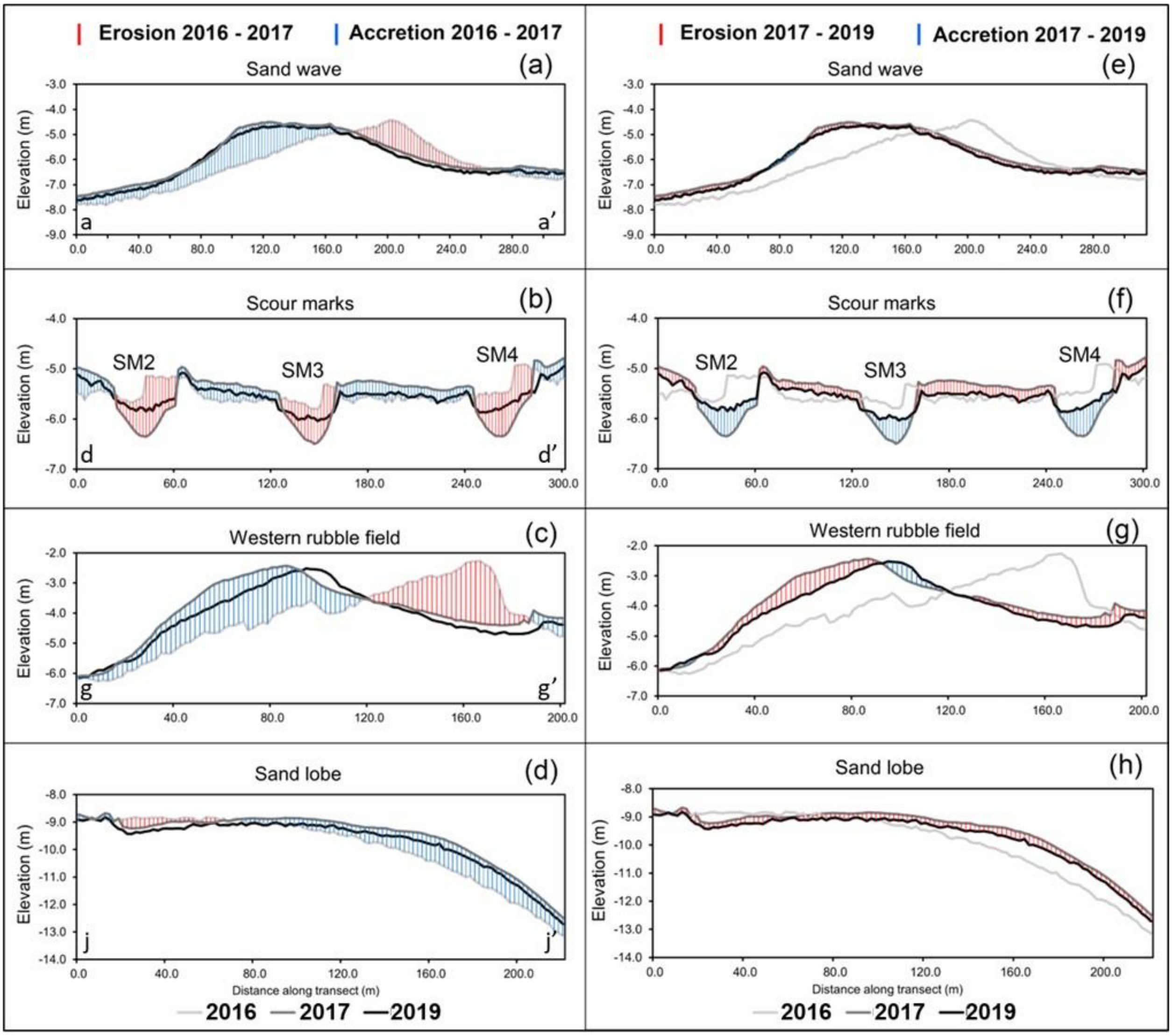

**Figure 7. Elevation transects across geomorphic features in 2016, 2017, and 2019.** Geomorphic features included a sand wave (a, e), scour marks (b, f), western reef rubble field (c, g), and a sand lobe (d, h). Lowercase letters indicate direction of transects as shown in Figure 6. Vertical red lines indicate areas of erosion and vertical blue lines indicate areas of accretion between (a-d) 2016 and 2017 (before and after Hurricane Irma) and between (e-f) 2017 and 2019. SM = scour mark.

### 3.2.2 Scour Marks

Development of scour marks was observed in seagrass and unconsolidated sediment habitats in the back reef area of Looe Key Reef between 2016 and 2017 (Fig. 6d, e, and f). These features ranged from approximately 30 to 60 m in length and width with average depths of approximately 5.7 to 7.5 m in 2017. Visual validation of select scour features indicated they developed

between 2016 and 2017 at the edges of seagrass beds where small (approximately 0.5 m) ledges marked the transition between the slightly higher elevation of seagrass beds and lower elevation of adjacent unconsolidated sediment (Fig. 6f). Transect analyses showed considerable erosion of the western boundaries of seagrass beds, development of pit-like features up to approximately 20 m in diameter and 1 m deep, transport of sediment westward, and burial of seagrass between scour features (Fig. 7b). Scour marks showed some infilling between 2017 and 2019. Validation imagery showed exposed rhizomatous growth at the western edges of seagrass beds (Fig. 8).

Elevation- and volume-change analyses were performed on four examples of these features (Fig. 6d and e). Scour mark 1 was 714 $m^2$ with 99% of the area consisting of discontinuous seagrass. Between 2016 and 2017, mean elevation change was -0.49 m (Table 2) with a maximum observed change of -1.09 m. Net volume change was less than -0.001 $Mm^3$, and area-normalized volume change was approximately -0.51 $Mm^3$ $km^{-2}$ (Table 3). Between 2017 and 2019, this feature showed accretion with mean elevation change of 0.10 m and a net volume change of less than 0.001 $Mm^3$ (Tables 2 and 4). Area-normalized volume change was approximately 0.12 $Mm^3$ $km^{-2}$. Scour mark 2 was 1,400 $m^2$ with 88% of the area consisting of continuous seagrass and 12% unconsolidated sediment. Between 2016 and 2017, mean elevation change was -0.50 m (Table 2) with maximum observed change of -1.28 m. Net volume change was less than -0.001 $Mm^3$ and area-normalized volume change was approximately -0.53 $Mm^3$ $km^{-2}$ (Table 3). Ninety-four percent of net volume change was associated with continuous seagrass habitat, which also had the highest area-normalized volume change of -0.56 $Mm^3$ $km^{-2}$. Between 2017 and 2019, this feature showed accretion with mean elevation change of 0.20 m and net volume change of 0.0003 $Mm^3$ (Tables 2 and 4). Continuous seagrass showed an increase in mean elevation (0.24 m) and net volume (0.0003 $Mm^3$) while unconsolidated sediment showed a decrease in mean elevation (-0.05 m) and net volume (less than -0.0001 $Mm^3$). Area-normalized volume change across the entire scour mark was approximately 0.22 $Mm^3$ $km^{-2}$. Scour mark 3 was 1,882 $m^2$ with 100% of the area consisting of continuous seagrass. Between 2016 and 2017, mean elevation change was -0.50 m with a maximum observed change of -1.25 m (Table 2). Net volume change was -0.001 $Mm^3$ and area-normalized volume change was approximately -0.52 $Mm^3$ $km^{-2}$ (Table 3). Between 2017 to 2019, this feature showed accretion with mean elevation change of 0.12 m and net volume change of 0.0003 $Mm^3$ (Tables 2 and 4). Area-normalized volume change was approximately 0.14 $Mm^3$ $km^{-2}$. Scour mark 4 was 1,520 $m^2$ with 99% of area consisting of continuous seagrass and 1% unconsolidated sediment. Between 2016 and 2017, mean elevation change was -0.54 m with a maximum observed change of -1.29 m (Table 2). Net volume change was -0.0009 $Mm^3$ and area-normalized volume change was approximately -0.57 $Mm^3$ $km^{-2}$ (Table 3). Ninety-nine percent of net volume change was associated with continuous seagrass habitat which also had the highest area-normalized volume change of -0.57 $Mm^3$ $km^{-2}$. Between 2017 to 2019, this feature showed accretion with mean elevation change of 0.12 m and net volume change of -0.0002 $Mm^3$ (Tables 2 and 4). Area-normalized volume change was approximately 0.13 $Mm^3$ $km^{-2}$. More than 99% of net volume change was associated with continuous seagrass habitat which also had the highest area-normalized volume change of approximately 0.14 $Mm^3$ $km^{-2}$.

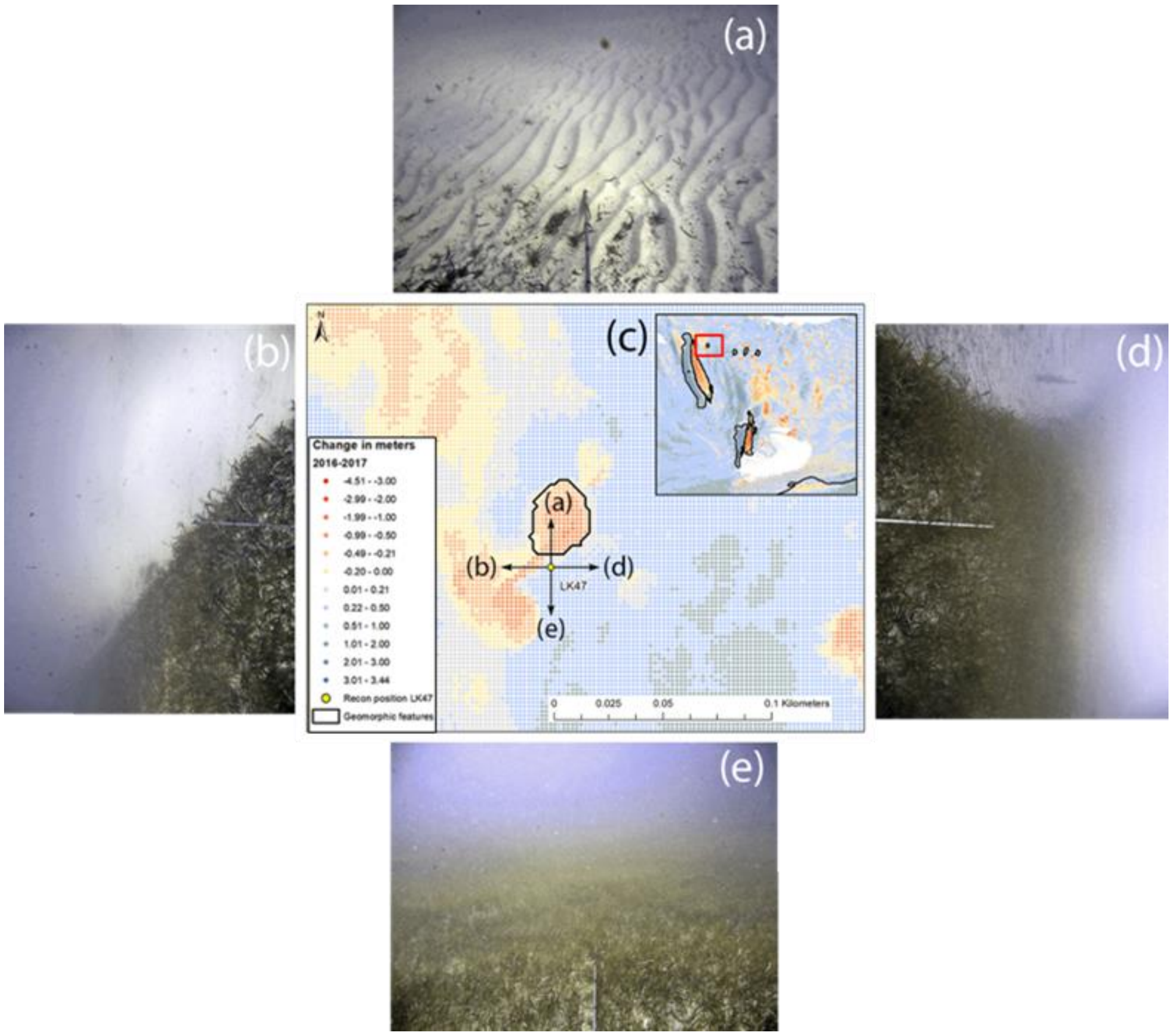

**Figure 8. Cardinal orientation imagery (a, b, d, and e represent north, west, east, and south, respectively) and elevation change (c) at a scour mark location used to validate benthic features observed in elevation change data.** East and west arrows show the boundaries between seagrass beds and sand flats in the elevation change data (c) and imagery (b and d). High erosion was noted between 2016 and 2017 on the sand flat (western) side of the habitat transition and minimal accretion was noted on the seagrass bed (eastern) side of the habitat transition. Photo credit: Mitch Lemon, Cherokee Nations System Solutions for U.S. Geological Survey.

### 3.2.3 Rubble Fields

Migration of reef rubble fields was observed in areas north and northeast of Looe Key Reef between 2016 and 2017. The largest of these features was approximately 418 m long and x 122 m wide at its widest point in 2017, 3 m in height from the crest to base on the deepest (western) edge, with average water depth of approximately 3.3 m (Fig 6g, h, and i). Transect elevation profiles showed the location of this feature in 2016, westward migration of approximately 80 m (crest to crest) in 2017, and minor eastward migration of 8 m (crest to crest) in 2019 (Fig. 7c). The accretion area of this feature covered an area of about 0.03 km$^2$ including approximately 43% reef rubble, 49% discontinuous seagrass, and 9% unconsolidated sediment. Between 2016 and 2017, mean elevation change of the accretion area (2017 location of the rubble field) was 0.89 m (Table 2) with a maximum elevation gain of 2.21 m. Total net volume change was 0.025 Mm$^3$ and area-normalized volume change was

0.914 $Mm^3$ $km^{-2}$ with discontinuous seagrass accounting for 54% of net volume change indicating burial of seagrass during migration of the rubble field (Table 3). Highest mean elevation and area-normalized volume changes were also associated with discontinuous seagrass habitat. An area of erosion (0.023 $km^2$) was observed in 2017 at the original 2016 location of the rubble field that was approximately 428 m long and x 78 m wide including 58% reef rubble, 31% discontinuous seagrass, and 11% unconsolidated sediment. Mean elevation-change of the erosion area between 2016 and 2017 was -0.63 m (Table 2) with a maximum elevation loss of -2.11 m. Total net volume change was approximately -0.015 $Mm^3$ and area-normalized volume change was -0.661 $Mm^3$ $km^{-2}$ with 69% of net volume change associated with reef rubble (Table 3). Highest mean elevation and area-normalized volume changes were also associated with reef rubble.

Between 2017 and 2019, the rubble field (accretion area) showed mean elevation change of -0.24 m, net volume change of -0.007 $Mm^3$, and area-normalized volume change of -0.248 $Mm^3$ $km^{-2}$ (Tables 2 and 4). Discontinuous seagrass showed greatest loss in mean elevation and area-normalized volume change and accounted for 59% of net volume change. The adjacent erosion area (original 2016 location of the rubble field) showed a mean elevation change of -0.10 m (Table 2) with a maximum elevation loss of -0.52 m. Total net volume change was approximately -0.002 $Mm^3$ and area-normalized volume change was -0.103 $Mm^3$ $km^{-2}$ with 53% of net volume change associated with discontinuous seagrass (Table 4). Highest mean elevation and area-normalized volume changes were also associated with discontinuous seagrass. Mean elevation and volume losses generally decreased with increasing mean habitat depth in the erosion area (Tables 2 and 4).

**3.2.4 Sand Lobe**

Substantial accretion was observed along a sand lobe located near the base of the fore-reef slope of Looe Key Reef between 2016 and 2017 (Fig. 6j, k, and l). This feature was approximately 1,383 m long and 344 m wide (approximately 0.27 $km^2$) at the widest point with an average water depth of approximately 11.9 m in 2017 and included only unconsolidated sediment habitat. Between 2016 and 2017, mean elevation change was 0.51 m (Table 2) with maximum gains in elevation up to 1.5 m along the southern (seaward) downslope section of this feature and maximum elevation losses of -0.58 m along the northern landward section, nearest to the base of the of the fore-reef slope (Fig. 7d). Total net volume change was 0.14 $Mm^3$ and area-normalized volume change was 0.51 $Mm^3$ $km^{-2}$ (Table 3). Between 2017and 2019, mean elevation change was -0.20 m with maximum elevation losses up to -1.12 m (Table 2, Fig. 7d). Only 852 of 67,389 elevation points analysed for this feature showed gains in elevation after 2017, averaging 0.05 m. Transect elevation profiles showed relatively consistent losses in elevation (erosion) across the sand lobe north to south (landward to seaward) during this time-period. Total net volume change was -0.05 $Mm^3$ and area-normalized volume change was -0.20 $Mm^3$ $km^{-2}$ (Table 4).

## 4 Discussion

There are few comprehensive assessments of the effects of major hurricanes on seafloor elevation and geomorphology on coral reefs; and no quantitative studies of reef-scale seafloor elevation change resulting from tropical storm impacts have previously been conducted in the Florida Keys. Our results showed Hurricane Irma was primarily a depositional event that increased mean seafloor elevation and volume over a 15.98 km$^2$ section of Looe Key Reef by 0.34 m (annualized elevation-change rate of up to 247 mm yr$^{-1}$) and up to 5.4 Mm$^3$, respectively, with area-normalized volume change of approximately 0.34 Mm$^3$km$^{-2}$. Our observations were based on elevation measurements collected 13.5 months before the storm and three to six months after the storm and, therefore, included any persistent change that occurred during quiescent sea state conditions before and after the passing of Irma. However, observations during several rapid reef assessments after the storm indicated broad-scale sediment deposition as a direct result of Hurricane Irma (Viehman et al., 2018; Walker, 2018; Wilson et al., 2020; Kobelt et al. 2019), which corroborates our findings of increased mean elevation and sediment accretion resulting from this storm event. Furthermore, wind conditions were relatively quiescent from the 2016 lidar acquisition date up to the passing of Hurricane Irma and after the storm, and historical aerial imagery of LKR from 2014 and 18 March 2017 (3 years and 6 months prior to Hurricane Irma, respectively, Fig. 9) show that patterns of major sedimentary features were mostly static (Finkl and Vollmer, 2017) in the few years prior to the storm. Our 2016 to 2017 elevation change results showed general movement of sediment and migration of major geomorphic features from ENE to WSW in shallow areas (ranging from approximately 2 to 5.5 m water depth in 2016) of the reef proper and back reef area, consistent with the direction of sustained, high magnitude winds during the passing of Hurricane Irma (Fig. 4; Fig. 6a, d, and g). For example, large sand waves and rubble fields (approximately 0.02 to 0.06 km$^2$ in area) migrated westward approximately 80 m (Fig. 6a and g) causing burial of seagrass habitat. Scour marks developed due to erosion of the western edges of seagrass beds and westward transport of sediment, causing burial of adjacent seagrass beds between scour marks (Fig. 6d). Numerical modelling of the impact of hurricane-induced wave-current interactions on the transport of material along the FRT during Hurricane Irma showed that wave radiation stress primarily affected particle transport trajectories during the passage of the hurricane (Dobbelaere et al., 2022). Additionally, wave energy dissipation occurred through depth-induced wave breaking and bottom dissipation at the shelf break and over the coral reefs. Furthermore, after the passage of the hurricane, suspended particles were transported northeastward by the Florida Current (Fig. 1d) and were advected (via Stokes drift) from the outer shelf to inshore for approximately 2 days (Dobbelaere et al., 2022).

Similar geomorphic seafloor changes have been documented for other category 4 hurricanes in the Florida Keys based on photographic air and ground surveys, maps, sediment cores, and bottom markers. In 1967, Hurricane Donna approached from the southeast and passed over the central islands of the Florida Keys in September 1960 with sustained winds of 226 km h$^{-1}$ (category 4) and with breaking waves and storm currents causing broken coral rubble up to a meter in diameter, shoreward transport of gravel to boulder sized rubble and sand approximately 60 to 150 m shoreward, and burial of seagrass with 15 cm

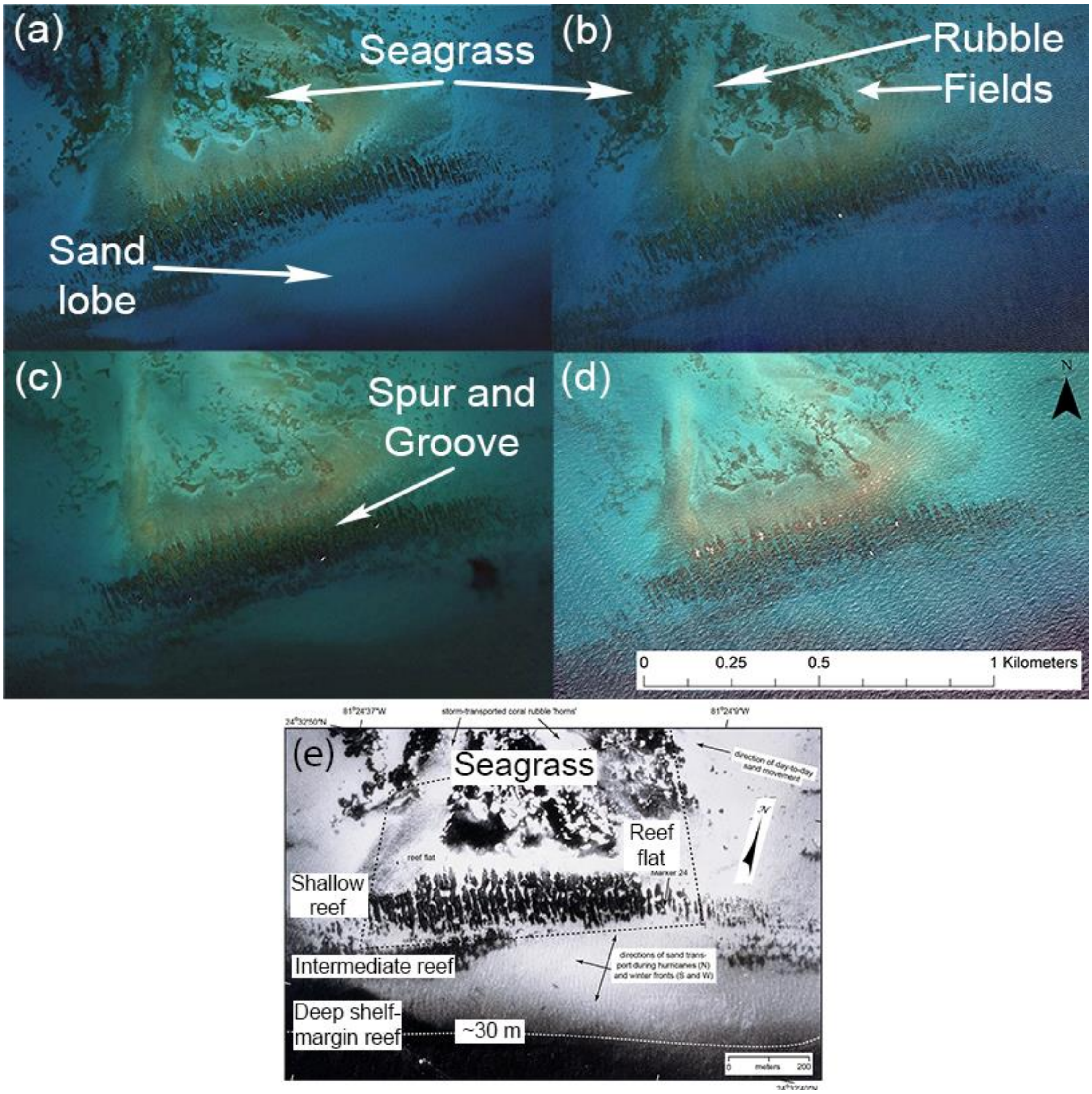

**Figure 9. Historical satellite and aerial imagery of Looe Key Reef.** Imagery from (a) 17 December 2014, before Hurricane Irma; (b) 18 March 2017, before Hurricane Irma; (c) 30 December 2017, 3 months after Hurricane Irma; (d) 2019, 16.5 months after Hurricane Irma; and (e) from 1975 (Lidz et al., 2016). Panels a, b, and c source: Maxar 2023 via © Google Earth Pro 7.3.6.9345, downloaded 11 September 2023. Panel d source: 2019 NOAA National Geodetic Survey via NOAA Digital Coast, downloaded 11 September 2023, https://www.fisheries.noaa.gov/inport/item/63292.

of sediment (Ball et al., 1967). Hurricane Betsy approached from the west and passed over the Florida Keys approximately 25 km north of Hurricane Donna's landfall in September 1965 with sustained winds of up to 226 km $h^{-1}$. While both storms had similar destructive effects to corals on the outer reefs, Hurricane Betsy produced less rubble, showed an overall effect of erosion and recycling of sediment in the environment, and caused sediment plumes from the mainland to the edge of the Gulf

Stream for several days after the Hurricane passed (Perkins and Enos, 1968). Perkins and Enos (1968) noted the difference in wind directions for the two storms caused different effects, and that it is difficult to extrapolate quantitative sedimentation rates from the sedimentary record of one hurricane and frequency of recorded hurricanes. Hurricane Andrew made landfall along the southeast coast of Florida just south of Miami also with sustained winds of 226 km $h^{-1}$ with maximum wave heights of less than 2 m. Branching corals were broken, massive coral heads were toppled, seafans and sponges were ripped loose, and shallow reefs sustained the most damage (Orr and Ogden, 1992); however, there was little damage to seagrass beds immediately seaward of coastal mangroves (Tilmant et al., 1994). Hurricane Georges was a category 2 storm that passed over Key West with maximum sustained winds of only 145 km $h^{-1}$. However, data from 30 seagrass monitoring transects showed a 3% decline in density of *Thalassia testudinum* and 19% decline in density of *Syringodium filiforme* seagrasses, with complete loss of seagrass beds at 3 monitoring stations, burial of one station with 50cm of sediment, substantial erosion at two stations (Fourqurean and Rutten, 2004). Fourqurean and Rutten (2004) showed that seagrass recovery was slowest at sites that were eroded; losses by mechanical thinning and burial with only a few centimeters of sediment recovered quickly; and seagrass buried with 10s of centimeters of sediment hadn't recovered by three years after the storm. Results from these studies show the variability in storm impacts due complex interactions among factors such as location, fetch, wind speed, duration, storm history, and water depth (Fourqurean and Rutten, 2004), and demonstrate the value of comprehensive, quantitative post storm assessments of geological and ecological impacts.

A previous analysis of seafloor elevation change at LKR during the decade prior to Hurricane Irma (from 2004–2016, during which only one minor tropical storm impacted this location in 2008) indicated an increase in mean elevation of 0.39 m (annualized elevation-change rate of 32.5 mm $yr^{-1}$), net volume gain of up to 6.4 $Mm^3$ and area-normalized volume change of 0.39 $Mm^3$ $km^{-2}$, with accretion observed across all habitat types and some WSW movement of sand waves (Yates et al., 2019). Our results showed that sediment deposited during the approximately 16.5 to 19.5-month time-period including impacts from Hurricane Irma caused changes in seafloor elevation and volume across all habitat types similar in magnitude to net changes observed over the past decade and at accumulation rates one order of magnitude greater. Previous studies on several coral reefs around St. Croix, U.S. Virgin Islands showed that physical transport of sediment is primarily due to wave-induced oscillatory and unidirectional currents, and that storms can increase sediment transport by an order of magnitude higher than during non-storm conditions (Hubbard et al., 1981; Hubbard, 1986). Measurements from 15 locations around St. Croix showed sediment transport rates ranging from 0.009 to 0.3 $Mm^3$ $km^{-2}yr^{-1}$ during non-storm conditions, and 0.09 to 1.5 $Mm^3$ $km^{-2}yr^{-1}$ during storm conditions (Hubbard et al., 1981; Yates et al., 2017). Sediment trap studies along the southwest coast of Puerto Rico showed median sediment accumulation rates increased by an order of magnitude (from approximately 6 to 68 mg $m^{-2}$ $d^{-1}$) after the passage of Hurricane Maria in September of 2017 (a category 4 storm) and a large October 2017 storm that caused resuspension of bottom sediments (Takesue et al., 2021). Furthermore, these accumulation rates exceeded the threshold of 10 mg $m^{-2}$ $d^{-1}$

which is considered heavy sedimentation and has been associated with fewer coral species, less live coral, lower coral growth rates, reduced coral recruitment and calcification rates, and slower rates of reef accretion (Rogers, 1990).

Mean elevation- and area-normalized volume-change from 2016–2017 for habitats examined in our study increased significantly with water depth suggesting that, in addition to broad-scale sediment deposition across the study site, sediment was also transported from shallower to deeper habitats (Fig. 5a and b). Notably, greatest increases in elevation (accretion) were associated with habitats in water depths exceeding 11 m including aggregate reef, a sand lobe consisting of unconsolidated sediment, and 'not classified' habitat located seaward and near the base of the reef's spur-and-groove formation, suggesting some movement of sediment offshore and downslope (Fig. 4a, Table 2). Additionally, erosion was observed in the shallower, upslope grooves of the spur-and-groove formation, and accretion was observed in the deeper, downslope areas of the grooves from 2016 to 2017 further suggesting downslope, offshore movement of sediments (Fig. 10). The sand lobe at the base of the spur and groove formation also showed upslope erosion and considerable downslope (seaward) accretion, further suggesting offshore transport of sediments (Fig. 7d). Our observations are consistent with previous bathymetric change analyses conducted along the northern FRT from 2001 to 2008 (approximately 3 years before Hurricane Ivan and 3 years after Hurricane Katrina) that showed movement of up to 1.8 $Mm^3$ of sediment between these time periods and transport of sediment from the inner shelf to offshore and beyond the shelf edge through gaps in the barrier reef and diabathic (cross-shore) channels during high-energy events or when the back reef overfills with sand (Finkl, 2004; Finkl and Vollmer, 2017). These observations are also consistent with results of Yates et al. (2017) that show a multi-decadal trend along the FRT of reef sediment transport down the fore-reef-slope and export offshore. Field observations of currents, waves, and reef sediment grain-size analyses coupled with integrated ocean-atmosphere-wave-sediment transport modelling during a one-year study at Crocker reef in the Upper Florida Keys showed that sediment mobility was primarily driven by wave stress exceeding critical shear stress; current stress alone only exceeded the critical shear stress for sediment mobility 5% of the time usually due to Florida Current eddies (Torres-Garcia et al., 2018). Torres-Garcia (2018) showed that nonbreaking wave stress (characteristic of quiescent sea states) mobilizes sand approximately 23 to 59% of the time; and fine-grained material is winnowed from the shallow areas of the reef and deposited to the flanks and offshore, particularly to the southwest. Furthermore, the critical stress threshold of gravel-sized material was exceeded only 1 to 13% of the time, particularly during near-field tropical storm conditions (similar to Hurricane Wilma, a category 3 hurricane) that cause breaking waves, mobilize and transport gravel material, and can cause physical reef degradation (Torres-Garcia et al., 2018). Southwest counter currents due to the formation of Florida Current eddies (Lee and Williams, 1999) and WSW movement of sand wave features over a decadal time-period (Yates et al., 2019b) have also been observed near LKR. Results from these previous studies suggest that some sediment transport observed in our study could be due to persistent transport of sand during quiescent sea state conditions; however, the large volume of material transported (including gravel-sized and larger reef rubble) during the short time-period of our study from 2016 to 2017 was likely due primarily to storm conditions caused by Hurricane Irma.

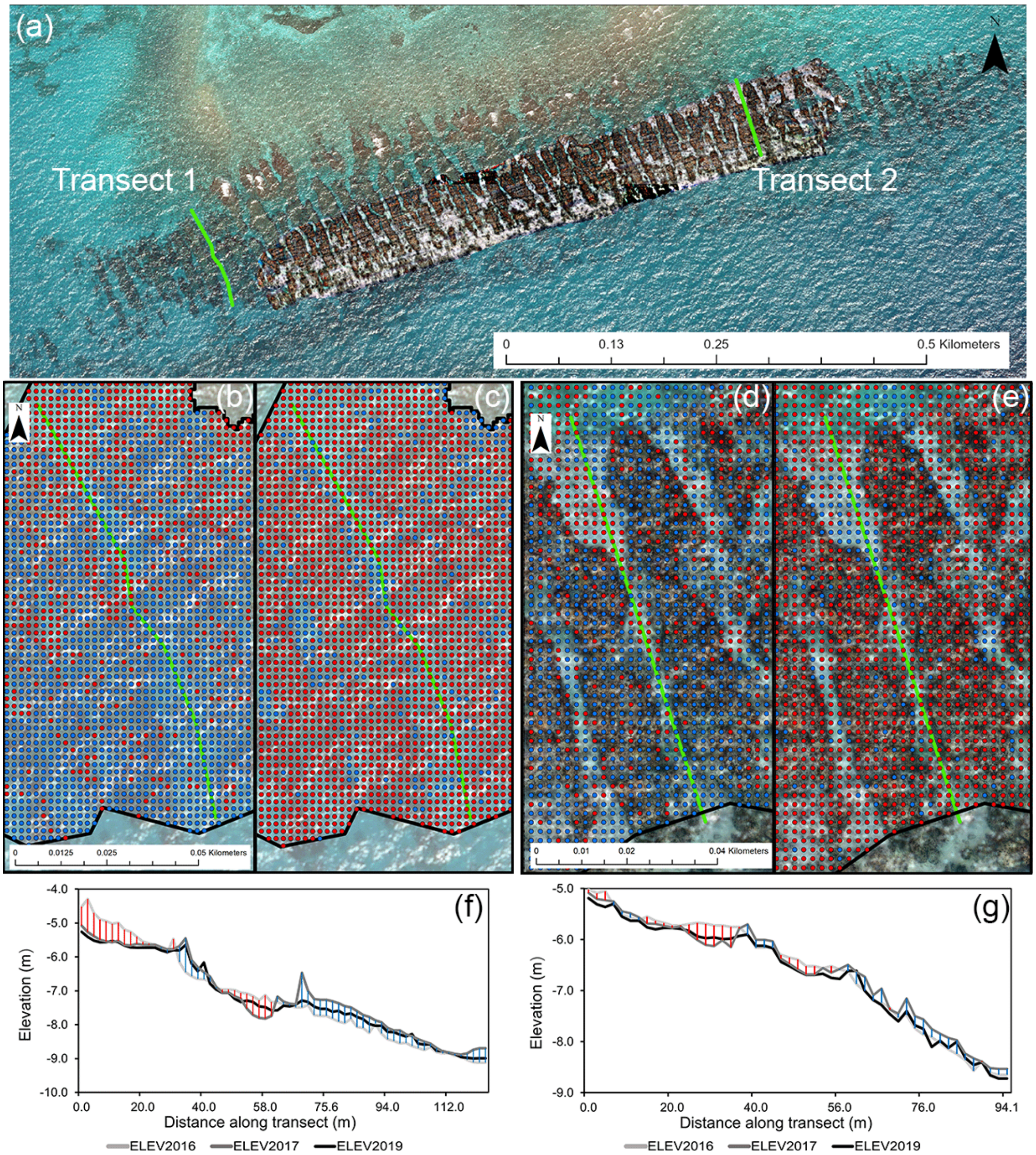

**Figure 10. Elevation-change along Looe Key Reef spur and groove formation.** (a) Upslope to downslope transects along Looe Key Reef spur and groove formation (green lines); image source: 2019 NOAA National Geodetic Survey via NOAA Digital Coast, downloaded 11 September 2023, [https://www.fisheries.noaa.gov/inport/item/63292](https://www.fisheries.noaa.gov/inport/item/63292) with structure from motion overlay image of Hatcher et al. (2022). Areas of erosion (red circles) and accretion (blue circles) along transect 1 between (b) 2016 and 2017, and between (c) 2017 and 2019. Areas of erosion (red circles) and accretion (blue circles) along transect 2 between (d) 2016 and 2017, and between (e) 2017 and 2019. Elevation profiles from 2016, 2017, and 2019 for (f) transect 1 and (g) transect 2. Vertical red lines indicate net erosion and vertical blue lines indicate net accretion between 2016 and 2017.

Approximately 16.5 months after Hurricane Irma (during a 13-month period between 2017 and 2019), net erosion was observed across all habitats with mean elevation-change of -0.15 (annualized elevation change-rate of -139 mm $yr^{-1}$), net volume change up to -2.46 $Mm^3$, and area-normalized volume change of -0.15 $Mm^3km^{-2}$. Newly deposited carbonate sediments typically have porosities of 40 to 70% (Choquette and Pray, 1970) at shallow sediment depths of a few hundreds of meters (Schmoker and Halley, 1982). Porosity of carbonate sands on the FRT and in Hawk Channel ranges from 60 to 72% in the upper 22 cm of deposited sediment (Walter et al. 2007). Schmoker and Halley (1982) showed that there is little or no sediment porosity loss at near-surface sediment depths. Application of their exponential function for porosity versus depth of sediment (porosity (%) = $41.73e^{-z/2498}$, where z = depth below sediment surface) indicates that the decrease in porosity of deposited carbonate sediments at 2 m below the sediment-surface is only 0.03%. Carbonate sands have settling velocities ranging from 0.025 to 0.364 m $s^{-1}$ (Riazi et al., 2020). Satellite imagery shows the sediment plume caused by resuspension of sediment during Hurricane Irma cleared within approximately 5 days of the storm's passing (NASA, 2023). Therefore, it is likely that resuspended sediment settled quickly (within days) when storm conditions subsided; and it is unlikely that the decrease in elevation observed between 2017 and 2019 was caused by compaction of sediment after the storm. This suggests that approximately 50% of sediment deposited between 2016 and 2017 was eroded by 2019 due to physical transport away from the study site. The sand wave and reef rubble field showed continued erosion between 2017 to 2019 with some evidence for migration of the crest of the rubble field back toward its original 2016 position indicated in the elevation profile (Fig. 6 and 7). Shallow areas between the scour marks showed erosion, while the scour mark pits showed infilling (Fig. 6 and 7). Spurs of the spur-and-groove formation primarily showed erosion, while shallow (landward) sections of grooves showed some accretion, likely due to transport of sediments from spurs to grooves and downslope from the shallow reef (Fig. 10). Deeper (seaward) areas of grooves and the sand lobe located at the base of the spur and groove formation showed erosion (Fig. 4 and 6k) suggesting continued downslope, offshore transport of sediments. Historical aerial and satellite imagery from before and after the passing of Hurricane Irma corroborates our elevation-change observations (Fig. 9). Imagery from 2014 and March 2017 shows that major geomorphic features of Looe Key proper such as distribution of seagrass beds and the size and position of the sand lobe and rubble fields were relatively static between these time periods leading up to Hurricane Irma (Fig. 9a and b). Imagery from December 2017, 3 months after Hurricane Irma passed, shows broad scale sediment deposition and burial of seagrass beds in the shallow areas of the reef proper, erosion and exposure of deeper, downslope spur-and-groove formation and downslope deposition on the sand lobe (Fig. 9c). Imagery from 2019 shows re-exposure of some shallow seagrass beds and deep spur-and-groove formation as sediments were eroded (Fig. 9d). Historical areal imagery from 1975 (Fig. 9e, Lidz et al., 2016) shows a distribution of seagrass, presence of rubble fields, and patterns of sediment along the sand lobe similar to 2014 and 2017 imagery (before Hurricane Irma) indicating these features have persisted over the past several decades despite repeated impact from tropical and seasonal storms. Lidz et al. (2016) suggested the formation of rubble fields in the shallow back reef area is mainly due to historical passage of hurricanes and winter storms, and our elevation change results suggest that these structures continue to migrate in response to storm conditions. Lidz et al. (2016) also suggested that transport of sediment during hurricanes was

primarily to the north; however, our observations showed primary sediment movement during Hurricane Irma was WSW and downslope from shallow to deep habitats with apparent seaward movement of the sand lobe after the storm passed.

Previous examination of multi-decadal elevation-change in a 19 km$^2$ study site at Looe Key Reef from 1938 to 2004 showed mean elevation change of -0.30 m (annualized elevation-change rate of -4.5 mm yr$^{-1}$), net volume loss up to -5.7 Mm$^3$, and area-normalized volume change of -0.30 Mm$^3$ km$^{-2}$ indicating a long-term trend of erosion at this location over more than six decades (Yates et al., 2017). Similar results were observed for a 241 km$^2$ area of the Upper Florida Keys with an annualized elevation-change rate of -1.4 mm yr$^{-1}$ between 1934 and 2004 (Yates et al., 2017). Furthermore, six of nine habitats at LKR showed elevation loss over those periods, with greatest losses associated with shallow habitats, and mean elevation and volume gains in deep-water habitats including at the base of the spur-and-groove habitat, indicating transport of reef sediments down the fore-reef-slope and export offshore (Yates et al., 2017). Our observed rate of mean elevation loss between 2017 and 2019 (-139 mm yr$^{-1}$) was two orders of magnitude higher than the multi-decadal rates of Yates et al. (2017). Additionally, elevation loss (erosion) showed a moderate correlation with water depth, and mean elevation losses during 2017 to 2019 were significantly greater in habitats with larger mean elevation gains during 2016 to 2017, suggesting that sediment distribution was re-equilibrating or stabilizing to quiescent sea-state conditions up to 16.5 months after the storm.

The annualized mean rate of elevation-change for LKR from the 2.5-year period between July 2016 to January 2019 examined in our study, including sediment accretion from Hurricane Irma and the post-storm erosion and re-equilibration, was approximately 72 mm yr$^{-1}$, which is almost double the rate of accretion observed in the previous decade of 32.5 mm yr$^{-1}$ (Yates et al. 2019b). Numerous field reconnaissance observations immediately after the passing of Hurricane Irma indicated broadscale sediment deposition across the FRT due to the storm (e.g., Viehman et al., 2018; Walker, 2018; Wilson et al., 2020; Kobelt et al. 2019). Our 2016 to 2019 elevation-change rate is consistent with annualized mean elevation-change rates from 2016 to 2019 for the Lower FRT from approximately Big Pine Key to Key West of 84 mm yr$^{-1}$, and for the FRT from Miami to Key West of 76 mm yr$^{-1}$ (Fehr et al., 2021), further suggesting that sediment distribution may have still been undergoing post-storm re-equilibration at our study site and along the broader FRT (Table 5).

Collection and analysis of additional elevation-change data sets over shorter time-periods (e.g., seasonal to annual) could improve characterization of post-storm elevation-change rates and duration of post-storm sediment re-equilibration periods relative to persistent seasonal, interannual, decadal, and multi-decadal time periods. Our results also suggest that caution should be used in selection of DEMs for use in elevation change and projection modelling to minimize bias that could result from selecting elevation surfaces that reflect periods of rapid elevation change due to storm impacts and periods of post-storm re-equilibration.

**Table 5. Annualized mean elevation-change rates (mm yr$^{-1}$) for event-driven to multi-decadal time periods at the Florida Keys Reef Tract.**

| | Annualized mean elevation change rate (mm yr$^{-1}$) | | | | |
|---|---|---|---|---|---|
| **Location** | **Event-driven[a] 2016 to 2017** | **Post-storm[b] 2017 to 2019** | **Short-term[c] 2016 to 2019** | **Decadal 2004 to 2016** | **Multi-decadal 1930s to 2000s** |
| **Looe Key Reef** | 247 | -139 | 72 | 32.5[d] | -4.5[e] |
| **Lower Florida Reef Tract (south of Big Pine)** | na | na | 84[f] | na | na |
| **Florida Reef Tract (Miami to Key West)** | na | na | 76[f] | na | na |
| **Upper Florida Reef Tract (Elliott Key to Tavernier Key)** | na | na | na | na | -1.4[e] |

a = calculated assuming a total time-period of 16.5 months (13.5 month pre- to 3 months post-storm); b = total time period 13.5 months (3 to 16.5 months post-storm); c = total time-period approximately 30 months (13.5 months pre- to 16.5 months post-storm); d = using data from Yates et al., 2019; e = using data from Yates et al., 2017; f = using data from Fehr et al., 2021; na = no data available.

## 5 Conclusion

High-resolution lidar and multibeam bathymetric data were used to quantify seafloor elevation and volume change within the Looe Key Reef system of the Florida Keys Reef Tract over a 2.5-year period from 2016–2019 and to examine impacts from category-4 Hurricane Irma and post-storm re-equilibration of seafloor sediments. Analysis of seafloor elevation and volume change over a 16.5-month period from July 2016 to December 2017 showed Hurricane Irma caused broadscale deposition of sediments across all benthic habitats of this reef system and burial of seagrass and coral dominated habitat. Rates of net elevation change were one order of magnitude greater during this short-term period that included storm impacts from Hurricane Irma than for the previous decade (Yates et al., 2019). Major seafloor geomorphic features such as sand waves and rubble fields migrated 10s of meters to the WSW in response to predominant wind conditions during the passing of Hurricane Irma, and sediment accretion was significantly greater in deep habitats than shallow habitats, suggesting downslope and offshore transport of seafloor sediment.

Loss of mean elevation and volume in all habitats in the period following the storm (from December 2017 to January 2019) indicated that 35% to 50% of sediment deposited during the storm had eroded by approximately 16.5 months after the storm and that erosion rates were two orders of magnitude greater than historical, multi-decadal rates of erosion. Sediment erosion after the storm (2017–2019) was moderately correlated with depth and was significantly greater in habitats that showed greater accumulation during the period including Hurricane Irma from 2016–2017, suggesting a period of rapid sediment re-equilibration after the storm. Historical satellite and aerial imagery show that major geomorphic features at this location including rubble fields, sand waves, and a sand lobe at the base of the spur-and-groove formation have persisted over the past several decades despite impacts from storms. However, our elevation-change results indicate these features are highly ephemeral, migrating rapidly during storms, re-equilibrating to non-storm sea state conditions between storms, and periodically burying seafloor habitat such as seagrass. Such features and the area surrounding them likely represent localized areas of long- and short-term seafloor instability that could be less suitable for restoration of slow growing benthic species. Our observed

rates of elevation change in the 16-month period after Hurricane Irma were one to two orders of magnitude greater than during the past decade or multi-decadal period (Yates et al., 2017; 2019b) indicating seafloor sediments across all habitats may have still been re-equilibrating to non-storm sea state conditions up January 2019. Higher resolution elevation-change data collected over seasonal and annual time periods could improve characterization and understanding of short-term (event-driven, seasonal, interannual) and long-term (decadal to multi-decadal) rates and processes of seafloor change and help guide benthic habitat post-storm recovery and restoration efforts in topographically complex coral reef systems.

**Code availability**

Python script for the Seafloor Elevation Change Analysis Tool (SECAT), intended to be applied in ArcMap or ArcGIS Pro, is publicly available as a U. S. Geological Survey software release, doi: 10.5066/P9D5UUZ0, https://www.usgs.gov/software/seafloor-elevation-change-analysis-tool.

**Data availability**

Elevation-change and multibeam bathymetric data are publicly available in U.S. Geological Survey Data Releases at https://doi.org/10.5066/P9CHC95D, https://doi.org/10.5066/P937LNZF, https://doi.org/10.5066/P9NXNX61, https://doi.org/10.5066/P9JTOOMB and https://doi.org/10.5066/P9P2V7L0. Lidar topobathymetric data are publicly available from the NOAA Office for Coastal Management at https://www.fisheries.noaa.gov/inport/item/63018 and https://www.fisheries.noaa.gov/inport/item/48373. Seafloor habitat data are publicly available from the Florida Fish and Wildlife Conservation Commission, Fish and Wildlife Research Institute at http://ocean.floridamarine.org/IntegratedReefMap/UnifiedReefTract.htm.

**Author contribution**

KY and DZ conceptualized the research, data acquisition approach, and methodology for analysis of seafloor elevation data. KY, ZF, and DZ performed formal analysis of data. DZ developed the SECAT software for statistical analysis of elevation- and volume-change data. KY and ZF developed data interpretations and prepared the original manuscript draft with contributions from SJ and DZ. All authors contributed to preparation of the final, published manuscript.

**Competing interests**

The authors declare that they have no conflict of interest.

**Disclaimer**

Any use of trade, firm, or product names is for descriptive purposes only and does not imply endorsement by the U.S. Government.

**Acknowledgement**

Funding for this study was provided by the U.S. Geological Survey, Coastal and Marine Hazards and Resources Program and by 2018 Hurricane and Wildfire Supplemental Funding provided to the U.S. Geological Survey from the Additional Supplemental Appropriations for Disaster Relief Requirements Act of 2018 (P.L. 115-123). We would like to thank J.J. Fredericks, B.J. Reynolds, and A.S. Farmer for collection of the multibeam bathymetry data and J.J. Fredericks for development of the associated multibeam digital elevation model. We also thank J. Zieg for assistance with development of the SECAT software and K. Murphy for assistance with development of methods for sub-sampling large-scale digital elevation models. We greatly appreciate reviews of the original draft manuscript and constructive comments from L. Toth and G. Hatcher.

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
