# Peer review of "Impact of Hurricane Irma on Coral Reef Sediment Redistribution at Looe Key Reef, Florida, USA"

_EGUsphere, 2023_

## Author Comment (AC2)

**AUTHOR RESPONSE TO REVIEWER 1**
(Author responses are located below each reviewer comment)

Reviewer General comments:
This is a very good paper based on a rare data set that is well processed, providing both original results and a very informative discussion. Processing is simple in its principles, but rigorous. This simplicity makes the paper easy to follow, even with a rich set of results.

The study has different scales of analysis and at the habitat-scale, it brings very interesting details. Also, some of the results could not really be foreseen (fig 5) and this brings a lot to the hurricane and coral reef literature, even if it specific to a Florida reef.

The reader will have access to detailed results that are not necessary to review here, but the study certainly provides a unique database in the context of Caribbean-Atlantic reefs, and as it claims, it fills some major gaps.

Really I don't see much to complain about with this paper, except two minor comments. It could be publish as is, which is something I have seen only twice in my reviewing career.

*Author Response*:
Very many thanks for your helpful review, perspectives, and comments. We've provided responses to your specific comments below and have revised the manuscript as indicated.

Reviewer Comment:
L122: I would not say Florida Keys, or the FRT in particular, is a barrier reef ; No way. Rather call them shelf reefs.

*Author Response:*
We appreciate the reviewer's comment and have changed this text to indicate 'shelf reef' instead of a 'barrier reef' to be consistent with literature on the geologic structure of the reef system (line 130 in revised manuscript). We have also included a reference to Lidz et al. (2003) for a more detailed discussion of the Florida Keys Reef Tract. The Florida Reef Tract is often referred to as a 'barrier reef' or 'barrier bank reef' by its management agencies but we agree that 'shelf reef' is a more appropriate term for this study (for example, the NOAA Florida Keys National Marine Sanctuary https://floridakeys.noaa.gov/corals/coralreef.html#:~:text=The%20Florida%20Keys%20are%20home,is%20about%20four%20miles%20wide, and Florida Department of Environmental Protection, Florida's Coral Reefs | Florida Department of Environmental Protection).

Lidz, B.H., Reich, C.D., and Shinn, E.A. 2003. Quaternary submarine geology in the Florida Keys. GSA Bulletin 115:845-866. https://doi.org/10.1130/0016-7606(2003)115<0845:RQSGIT>2.0.CO;2

Reviewer Comment:
L541 (and elsewhere): Accretion: I would not use this word here, as I would consider it as the process of incorporating the sediments and other calcareous material in the reef structures itself, hence a hardening process different than sediment deposition and movements.

*Author Response*:
We understand the reviewer's perspective, particularly with respect to hardening of materials deposited on and within reef structure that contributes to vertical accretion. However, we respectfully disagree that hardening, or lithification, is a requirement for defining accretion on coral reefs. For example, coral reef cores can often show accretion of unlithified sediments over short-term (seasons to years) to long-term (100s+ years) periods of time. Our elevation change analyses in this study represent change over short-term time periods (seasons to years) and our discussion includes comparisons to previous elevation change analyses over years to decades. These elevation change measurements reflect the net outcome of all processes that increase or decrease elevation (either gradually or due to storm events), including sediment deposition or erosion due to physical transport within the system, sediment creation (for example, due to coral breakdown/degradation, etc.) or carbonate dissolution within the system, export or influx, compaction, etc. We feel that the term 'deposition' doesn't accurately capture what is measured by our elevation change measurements; thus, we chose to use the word accretion. We have added a sentence to section 2.3, first paragraph (lines 201-203 in revised manuscript) to clarify that we are using elevation change to estimate the net effect of all processes affecting accretion and erosion in the system; and our use of these terms throughout this study indicates estimates of net accretion and net erosion from our measurements.

Reviewer Comment:
Then, may be no need to talk about restoration (L32, abstract). This is the trendy word of the moment in coral reef literature and beyond. Everyone is using it, which bothers me, and it is not justified here. Or add a specific technical paragraph in the Discussion in how there results could really help restoration (only one hint on restoration taking into account the results is provided in the conclusion).

*Author Response*:
We appreciate the reviewer's perspective and agree that a detailed technical discussion of application of our results to restoration is beyond the scope of our paper. We have removed the following text from the abstract (line 32) since it is not a primary focus of the manuscript: "and help guide benthic habitat post-storm recovery and restoration efforts in topographically complex coral reef systems". Our results do, however, provide valuable information on the physical state

of the seafloor environment with respect to rates of change in accretion and erosion that affect long and short-term seafloor stability. We feel this information could be applied to guide and improve placement of transplanted species on the seafloor to improve long-term restoration success (e.g., identification of locations that show long and short-term seafloor stability for placement of slow growing corals, or locations where placement of seagrass would be beneficial to reduce sediment erosion). Thus, we have chosen to keep the brief statement in the last paragraph of the conclusions that describes the potential utility of our results for this purpose.

Reviewer Comment:
Technical corrections: none, did not see any errors.

*Author Response*:
We appreciate your review.

**END OF REVIEW AND RESPONSE**

---

## Author Comment (AC3)

**AUTHOR RESPONSE TO REVIEWER 2**
(Author responses are located below each reviewer comment)

Reviewer General comments:
Here, we are given a welcome and specialized analysis of a particular reef within the Florida Reef Tract, with pre-storm, post-storm, and post-recovery LIDAR observations informing the transient dynamics associated with sedimentation incurred by hurricane stresses. While the results are mostly reasonable, the proposed analysis provides real-world evidence alongside data analysis that can inform future studies of highly transient sedimentary impacts on coral reefs.
The science is conducted very well, paper and analyses are very well written, and I only have minor comments to suggest in how the paper is framed, as well as a few grammatical/editing notes:

 *Author Response*:
Very many thanks for your helpful review, perspectives, and comments. We've provided responses to your specific comments below and have revised the manuscript as indicated.

Reviewer Comment:
L52: The discussion on seagrass here feels more of an afterthought, rather than given full appreciation of the dynamical role it plays in the ecosystem alongside corals. I suggest either elaborating upon it further, or leaving the discussion as an entirely physical analysis.

 *Author Response*:
We have added additional, relevant information to the introduction (lines 53-60 in revised manuscript) on long-term seagrass trends in South Florida including observed impacts from storm events. We have also expanded the discussion of storm impacts to seagrasses in the FKNMS including development of features similar to those observed in our study in Section 4.1, on lines 538-542 in the revised manuscript. We have added the following citations and references associated with this added text:

Carter, A. B., Collier, C., Coles, R., Lawrence, E., and Rasheed, M. A.: Community-specific "desired" states for seagrasses through cycles of loss and recovery, Journal of Environmental Management, 314, 115059. https:// doi. org/ 10. 1016/j. jenvm an. 2022. 115059, 2022.

Hastings, K., Hesp, P. and Kendrick, G. A.: Seagrass loss associated with boat moorings at Rottnest Island, Western Australia. Ocean & Coastal Management, 26, 225–246, https:// doi. org/ 10.1016/ 0964- 5691(95) 00012-Q, 1995.

Kirkman, H. and Kirkman, J.: Long-term seagrass meadow monitoring near Perth, Western Australia, Aquatic Botany, 67, 319–332, https:// doi. org/ 10. 1016/ S0304- 3770(00) 00097-8, 2000.

Krause, J. R., Lopes, C. C., Wilson, S. S., Boyer, J. N., Briceño, H. O., Fourqurean, J. W.: Status and trajectories of soft-bottom benthic communities of the South Florida Seascape revealed by 25 years of seagrass and water quality monitoring data, Estuar. Coast., 46, 477-493, https://doi.org/10.1007/s12237-022-01158-7, 2023.

Patriquin, D. G.: "Migration" of blowouts in seagrass beds at Barbados and Carriacou, West Indies, and its ecological and geological implications, Aquatic Botany, 1, 163–189, https:// doi. org/ 10. 1016/ 0304-3770(75) 90021-2, 1975.

Reviewer Comment:
L57-60: This sentence runs on and is hard to read.

*Author Response*:
We have modified the sentence on lines 57 to 60 (lines 64-67 in revised manuscript) as follows to improve readability:

"Continued FRT coral reef degradation and loss of seafloor elevation is projected to increase flooding risk from storms and coastal inundation to more than 7,300 people and to cause $823.6 million in direct and indirect damage to housing, buildings, and businesses, annually (Storlazzi et al., 2021, estimated in 2010 U.S. dollars, USD)."

Reviewer Comment:
L265, 270-271, 286: Units appear wrong - should be Mm instead of mM.

*Author Response*:
Thank you for catching this error. We have replaced all occurrences of 'mM' with 'Mm'. We have also found and corrected minor punctuation errors associated with some reference citations.

Reviewer Comment:
The first part of the discussion could also be heavily improved with more care taken towards structure and a central message. Often the messaging became diluted because focus would shift significantly through a paragraph. The final paragraphs are structured very well.

*Author Response*:
We have made several edits to restructure, better focus, and improve readability of the first part of the discussion. These changes are listed below. Please note that line numbers refer to the revised manuscript.

Section 4 Discussion: We have created two subsections in the discussion including section '4.1 Storm Impacts' and section '4.2 Post-storm Change' to clearly distinguish discussion elements related to each of these topics.

Section 4.1, paragraph 1, lines 483-486: The last sentence was modified as follows to clarify the primary message regarding Hurricane Irma as primarily a depositional event: "Historical aerial imagery of LKR from 2014 and 18 March 2017 (3 years and 6 months prior to Hurricane Irma, respectively, Fig. 9) shows that patterns of major sedimentary features were mostly static in the few years prior to the storm (Finkl and Vollmer, 2017), further suggesting that broad scale sediment deposition resulted directly from Hurricane Irma."

Section 4.1, paragraph 2, lines 488-505: Content of this paragraph was rearranged to improve organization of the information and transition from discussion of sediment deposition in paragraph 1 to the processes driving sediment transport in paragraph 2 as indicated in numerical modelling of hurricane impacts on hydrodynamics.

Section 4.1, paragraph 3: Minor edits were made to the first sentence (line 516) to clearly distinguish the change in topic discussion from causes of sediment transport, including movement of large geomorphic features identified in our study, to observations of similar hurricane impacts caused by other storms in the Florida Keys. Transition phrases were added to lines 517 and 535 to clearly identify information from other studies and supporting information relevant to recovery rates and examples from past storm impacts.

Section 4.1, paragraph 4: The annualized elevation-change rate from our results showing short-term hurricane impacts was added to line 553 to provide a more direct comparison to the magnitude of elevation change over the past decade. Text was rearranged on lines 554-557 and a transition phrase was added to line 559 to indicate comparison of similar results from previous studies on the magnitude of sediment associated with storm impacts in other geographic locations.

Section 4.1, paragraph 5: This paragraph (beginning on line 567) was divided into two separate paragraphs to separate the discussion of patterns of erosion from review of the underlying mechanisms (see reviewer comment regarding line 554 of the original manuscript) and author response below).

Section 4.1, paragraph 6: Text was rearranged in this paragraph to clearly introduce the main discussion of the potential for sediment mobility during quiescent sea state conditions versus during storm impact relative to our results and observations (lines 584-604).

Section 4.2, paragraph 1: Minor edits to text were made on lines 618, 620, 627, and 630-632 to better clarify the primary discussion focus of our observed post-storm elevation loss and evidence from previous studies supporting our conclusion that the elevation loss was due to erosion instead of sediment compaction.

Section 4.2, paragraph 2: Minor edits were made to line 635 (first sentence) to clearly indicated the transition of discussion to Geomorphic feature impacts.

Reviewer Comment:

L539-568: A break around L554 may be helpful, separating patterns of erosion and corresponding deposition from the review of the underlying mechanisms.

*Author Response*:

We have inserted a paragraph break at line 554 (now line 584 in the revised version) to separate the discussion of the underlying mechanisms of erosion. We have also added an introductory sentence to the paragraph and rearranged the discussion content to improve readability (lines 584-604).

Reviewer Comment:

L590 is an important point to make as it also validates potential concerns I would have regarding the reliability of LIDAR as a post-hurricane measurement device. However, the current placement of the note seems slightly out of place and suggests a much less pressing role in the ultimate analysis.

*Author Response*:

We agree collection of lidar data during the presence of post-storm sediment plumes drastically reduces the reliability of that data and analyses for which it is used. We included the statement on line 590 (quoted below) as evidence that the post-storm sediment plume cleared quickly (within 5 days) after the passing of the storm (well before post-storm bathymetry data were collected) and that carbonate sediments settled quickly (within days of the resuspension event). We would like to leave this sentence in its current location to support the discussion of post-storm elevation change in that paragraph. Hurricane Irma passed over the Florida Keys on 10 September 2017. Collection of the post-storm multibeam bathymetry data used in our study did not begin until 12 December 2017, three months after the storm passed. Furthermore, the post-storm multibeam data was collected by USGS specifically for this study and during known water quality/clarity conditions when no sediment plumes were present. Thus, the post-storm sediment plume did not affect the data we used in our study. Our analyses are dependent upon the accuracy and reported error of the published 2016 and 2019 lidar data sets; and the reported lidar error is applied in our calculation of error (RMSE) associated with elevation change (see section 2.3). We also performed a qualitative visual inspection of satellite imagery (NASA Worldview) that corresponds to the dates of lidar collection prior to application in our elevation change analyses to help identify environmental conditions (such as cloud cover, sediment plumes) that could affect data quality. While this provides no guarantee of lidar data quality, it assisted with selection of appropriate data sets for elevation change analyses to help reduce error.

From line 590 (lines 627-629 in the revised version): "Satellite imagery shows the sediment plume caused by resuspension of sediment during Hurricane Irma cleared within approximately 5 days of the storm's passing (NASA, 2023). Therefore, it is likely that resuspended sediment settled quickly (within days) when storm conditions subsided…"

Reviewer Comment:
L608 I believe this is a typo and should be 'aerial."

*Author Response*:
Thank you for catching this error. We have changed the word 'areal' to 'aerial' (now on line 648).

Reviewer Comment:
L613-615 This line that suggests conflict between the current work and a previous work feels unnecessary, because while the pattern of motion for major landforms may have been oriented WSW, at the larger scale, the primary source of sediment originally deposited onto LKR may have arrived from the south, and the following motion of the sand lobe was not a part of the during hurricane motion.

*Author Response*:
The intent of this text (copied below) was to illustrate the differences/variability in storm response, not to suggest conflicting results between the two studies. We have modified the sentence as indicated below to clarify that intent.

Original sentence from lines 613 to 615 (revised manuscript lines 653-657): "Lidz et al. (2016) also suggested that transport of sediment during hurricanes was primarily to the north; however, our observations showed primary sediment movement during Hurricane Irma was WSW and downslope from shallow to deep habitats with apparent seaward movement of the sand lobe after the storm passed.

Revised sentence: "Lidz et al. (2016) also suggested that transport of sediment during hurricanes was primarily to the north. Our observations showed primary sediment movement during Hurricane Irma was WSW and downslope from shallow to deep habitats with apparent seaward movement of the sand lobe after the storm passed, illustrating the variability in storm impacts associated with individual storm events."

**END OF REVIEW AND RESPONSE**